

# Suppression of scattering from slow to fast subsystems and application to resonantly Floquet-driven impurities in strongly interacting systems

**Friedrich Hübner⋆**

Physikalisches Institut, University of Bonn, Nussallee 12, 53115 Bonn, Germany

⋆ friedrich.huebner@kcl.ac.uk

## Abstract

We study solutions to the Lippmann-Schwinger equation in systems where a slow subsystem is coupled to a fast subsystem via an impurity. Such situations appear when a high-frequency Floquet-driven impurity is introduced into a low-energy system, but the driving frequency is at resonance with a high-energy band. In contrast to the case of resonant bulk driving, where the particles in the low-energy system are excited into the high-energy band, we surprisingly find that these excitations are suppressed for resonantly driven impurities. Still, the transmission through the impurity is strongly affected by the presence of the high-energy band in a universal way that does not depend on the details of the high-energy band. We apply our general result to two examples and show the suppression of excitations from the low-energy band into the high-energy band: a) bound pairs in a Fermi-Hubbard chain scattering at a driven impurity, which is at resonance with the Hubbard interaction and b) particles in a deep optical lattice described by the tight-binding approximation, which scatter at a driven impurity, whose driving frequency equals the band gap between the two lowest energy bands.

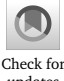

# 1 Introduction

In the past decades Floquet-engineering has become an important tool to manipulate quantum mechanical systems in the lab, in particular in cold atoms and materials in the context of quantum computing and quantum simulation [1–9]. It is based on the idea that external periodic driving of a system can be described by a time-averaged static Hamiltonian. In particular, this holds in the high-frequency regime where powerful analytical techniques, like the high-frequency expansion [2, 4, 10], are available to compute the effective Hamiltonians. Using Floquet-driving it is possible to design quantum systems which, beside many other applications, simulate novel phases of matter [11–13], allow to manipulate their transport properties [1,14], have non-trivial band structures [15–17] or couple to artificial gauge fields [4,18]. While traditionally the external drive is applied to the whole system (bulk driving), more recently there has been increased interest in driven impurities. Driven impurities can be used to Floquet-engineer effective impurities into systems. Such effective impurites can change the transport behaviour [19–22], exhibit interesting physics (such as Fano-resonances [23]) on their own, or they might be useful as devices in experiments, for instance as filters for particles with various properties [24–26].

    From a Floquet theory viewpoint, in order to implement a Floquet-engineered system in the high-frequency regime, the driving frequency $\omega$ should be as large as possible: the larger $\omega$, the slower the system will heat up due to the driving [27–29]. However, in practice, there is a limitation: the physical systems in the lab are often described by some low-energy effective description (like the tight-binding approximation), only the lowest band (or bands) is taken into account [30–32]. In reality, further high-energy bands also exist. In order to observe the correct effective Hamiltonian one must ensure that the driving frequency is not at reso-

nance with any of these high-energy bands, otherwise particles from the low-energy band are pumped into the neglected high-energy band. In the case of bulk driving, resonant driving leads to a hybridization of the low-energy band with the high-energy band, thereby effectively destroying the low-energy description, even for a large energy gap between both bands and weak driving [33]. Intuitively, the same should be true for a resonantly driven impurity: a particle from a low-energy band which scatters at the impurity should have a high chance of getting excited into the high-energy band. The contribution of this paper is to show that this intuition is wrong and contrarily establish the following surprising result: at a resonantly driven impurity, excitations from a low-energy band to a high-energy band are suppressed and are of order $\alpha = \frac{\text{bandwidth of low-energy band}}{\text{bandwidth of high-energy band}}$. Therefore, the low-energy description is still intact even if the driving frequency of the driven impurity is at resonance with some high-energy band. However, scattering inside the low-energy band at the impurity is strongly affected by the presence of the high-energy band.

This result will be based on a study of static scattering problems of the following peculiar type: imagine a system with two subsystems P and Q (each of which can contain multiple energy bands) coupled by a static impurity. We assume that particles in P evolve much slower than particles in Q (equivalently that Q's bandwidth is much larger than P's), while simultaneously both bands overlap (in order to allow scattering between them). This setting is quite unusual as typically a separation of time-scales also implies a separation of energy-scales. However, this situation naturally appears for resonantly driven impurities, since the system can overcome the energy gap by absorbing energy from the driving and thus both bands effectively overlap. We study scattering using the Lippmann-Schwinger formalism [34, 35] in the singular limit $\alpha = \frac{T_{\text{fast}}}{T_{\text{slow}}} = \frac{\text{bandwidth of low-energy band}}{\text{bandwidth of high-energy band}} \rightarrow 0$. We find that scattering from P to Q is of order $\mathcal{O}(\alpha)$ and therefore suppressed in the limit. The reason is that, due to the separation of time-scales, slow particles are not able to enter the impurity and therefore cannot get excited into Q. This manifests in an emergent artificial boundary condition for the scattering wavefunction: the scattering wavefunction vanishes at the location of the impurity. Still, scattering at the impurity can show non-trivial transmission and reflection even for $\alpha \rightarrow 0$ and can be evaluated explicitly. In fact, it is a universal result: the transmission through the impurity does not depend on the details of subsystem Q, but is completely determined by the low-energy subsystem P and the impurity. Since the results are fairly general and only require few physical assumptions, they apply to a broad range of physical systems showing the above mentioned separation of time-scales: the suppression of scattering from slow to fast systems will appear in interacting as well as non-interacting systems in any dimension, with any number of particles and is also independent of the driving strength of the impurity.

The generality of this result is crucial in order to apply it to driven impurities as well: since it only holds for static impurities one has to map the driven system onto an equivalent static one. This can be done using the extended Hilbert space formalism at the cost of a much increased Hilbert space [10, 36]. Nevertheless the static result is applicable. We also discuss two explicit examples: a) the breaking of bound pairs in an attractive Fermi-Hubbard chain at a resonantly driven impurity in the strong interaction limit $J \ll |U|$ (hopping parameter $J$, Hubbard interaction $U$) and b) the scattering of particles in an optical lattice at an impurity, which is driven at resonance with the energy gap between the two lowest bands. In both cases we show that the excitations from the lower band to the higher band are suppressed and also discuss the transmission through the impurity inside the lower band. These examples demonstrate the applicability of our results to two common low-energy approximation schemes, the Schrieffer-Wolff transformation and the tight-binding approximation. Beyond that, we expect that the results will apply generally to systems with low-energy descriptions or systems with a clear separation of slow and fast degrees of freedom (like spintronics systems [37, ]), given that they are suitably coupled by a resonantly driven impurity.

We believe that a result of this type can be valuable for studying resonantly driven impurities beyond the high-frequency limit: our analysis can be viewed as a zeroth order result of a systematic perturbative expansion in $\alpha$. If suitably established this perturbative expansion would also add another tool to the toolbox of Floquet theory. This could complement exact non-perturbative methods (like the non-equilibrium Green's function methods [21, 39–45]) in studying scattering at impurities and could perhaps be used to gain insights into impurities that are too computationally expensive to treat in practice via these methods. Most importantly, this expansion is non-perturbative in the impurity strength $\lambda$, which would allow to study impurities where the standard Born series for weak couplings [35] is not applicable.

The paper is structured as follows: in Section 2 we model systems with high and low-energy bands by means of the Schrieffer-Wolff transformation for strongly interacting systems. After adding the driven impurity, we find a peculiar physical situation which motivates the regime we analyse in this paper. In Section 3 we then introduce a simple exactly solvable toy model where the suppression of scattering becomes apparent. Based on this we then establish the general result for static systems in Section 4 and explain how to apply it to Floquet-systems in Section 5. In Appendix G and Appendix H we then demonstrate how to apply the results using two explicit examples: pair breaking in an attractive Fermi-Hubbard chain and excitations from the lowest to the second lowest band in an optical lattice, both at a resonantly driven impurity.

## 2 Motivation: Resonantly Floquet-driven impurities in strongly interacting systems

In this work we want to study scattering at a driven impurity, whose driving frequency $\omega$ is at resonance with the energy difference between two bands. Therefore, particles can get excited from one band into the other. The situation we have in mind is that we introduce a driven high-frequency impurity into some low-energy system $\tilde{P}$ (which might contain several energy bands). Due to the high-frequency driving the driven impurity might be at resonance with a band from a high-energy system $\tilde{Q}$, which does not affect the undriven system, due to the large energy difference between $\tilde{P}$ and $\tilde{Q}$.

In order to understand what happens in this scenario and to perform a mathematical analysis, it is important to physically model the high-energy system $\tilde{Q}$ and its relation to the low-energy system $\tilde{P}$. We will model this using a strongly interacting system in the large (attractive) coupling regime: in this regime typically a low-energy subspace $\tilde{P}$ emerges whose effective Hamiltonian can be computed using the Schrieffer-Wolff transformation [46, 47]. This gives rise to a particular separation of both time- and energy-scales in the system, which will be the starting point for our analysis.

We want to stress that this does not mean that our results only apply to strongly interacting systems: they are one way of generating a clear separation of scales and restricting to them will allow us to be more mathematical precise, especially when we want to compare driven impurities to bulk driving. An example for an emergent low-energy theory, which is not captured by the Schrieffer-Wolff transformation is the tight-binding approximation for lattice systems. There the relative scalings of the different systems are completely different, however our treatment will still apply to them, as we demonstrate in Appendix H.

Our work will apply to systems which have three ingredients. First, the uncoupled system is strongly interacting with an emergent low-energy subsystem. Second, a Floquet-drive is added to the system, whose driving frequency is at resonance with the energy gap between the low-energy subsystem and the high-energy subsystem. Third, the Floquet-drive is restricted to a localized region in space, i.e. the systems are coupled by a resonantly driven impurity, which will give rise to novel physics. We will now introduce these ingredients step by step.

## 2.1 Ingredient 1: Strongly interacting systems

Consider a strongly interacting quantum system with a Hamiltonian

$$\tilde{\mathbf{H}} = \tfrac{1}{\alpha}\mathbf{H}_{\text{int}} + \mathbf{H}_{\text{kin}}, \tag{1}$$

which can separated into two terms: the interaction $\mathbf{H}_{\text{int}}$ and the remaining part $\mathbf{H}_{\text{kin}}$ (we think of this part as the kinetic Hamiltonian, but in general it can contain more terms, like additional interactions, external potentials, etc.). The $\alpha \ll 1$ indicates that the interaction part is much stronger than the kinetic part (throughout this paper $\alpha$ will denote a small positive number). We are interested in studying the quantum system Eq. (1) in the regime $\alpha \to 0$. Mathematically the simplest way to understand this is to rescale time by a factor $\alpha$:

$$\mathbf{H} = \mathbf{H}_{\text{int}} + \alpha\mathbf{H}_{\text{kin}}, \tag{2}$$

and treat the kinetic part of the Hamiltonian $\mathbf{H}_{\text{kin}}$ as a perturbation to the interaction Hamiltonian $\mathbf{H}_{\text{int}}$. As an example consider the infinite attractive Fermi-Hubbard chain with lattice sites labelled by integers $n$:

$$\mathbf{H}_{\text{int}} = U\sum_n \mathbf{n}_{n\uparrow}\mathbf{n}_{n\downarrow}, \qquad \alpha\mathbf{H}_{\text{kin}} = -J\left[\sum_{n\sigma}\mathbf{c}_{n\sigma}^{\dagger}\mathbf{c}_{n+1\sigma} + \mathbf{c}_{n+1\sigma}^{\dagger}\mathbf{c}_{n\sigma}\right], \tag{3}$$

where $\mathbf{c}_{n\sigma}$ annihilates a fermion at site $n$ with spin $\sigma \in \{\uparrow, \downarrow\}$, $\mathbf{n}_{n\sigma} = \mathbf{c}_{n\sigma}^{\dagger}\mathbf{c}_{n\sigma}$, $J$ is the hopping parameter and $U = -J/\alpha < 0$ is the Hubbard interaction.

The Schrieffer-Wolff transformation [46, 47] is one method to systematically derive effective Hamiltonians in the regime $\alpha \to 0$. In the extreme limit $\alpha = 0$ the Hamiltonian simply becomes $\mathbf{H} = \mathbf{H}_{\text{int}}$, meaning that eigenspaces of $\mathbf{H}_{\text{int}}$ for different eigenenergies completely decouple. For simplicity, let us assume that $\mathbf{H}_{\text{int}}$ only has two eigenspaces with eigenvalues 0 and $E_{\text{exc}}$ (this happens for instance in the Fermi-Hubbard chain with two particles). The below arguments can be easily extended to multiple eigenspaces, see [47]. We denote the eigenspace for $E = 0$ by $\tilde{\mathsf{P}}$ (with is associated orthogonal projector $\tilde{\mathbf{P}}$) and the eigenspace for $E = E_{\text{exc}}$ by $\tilde{\mathsf{Q}}$ (with is associated orthogonal projector $\tilde{\mathbf{Q}}$). Throughout this paper we will denote subspaces in the physical (often driven) system with a tilde, in order to distinguish them from subsystems we define in our mathematical analysis on abstract systems, which are denoted without tilde. At first order in $\alpha$ the effective Hamiltonians in $\tilde{\mathsf{P}}$ and $\tilde{\mathsf{Q}}$ are simply given by [47]

$$\mathbf{H}^{\tilde{\mathsf{P}}} = \alpha\tilde{\mathbf{P}}\mathbf{H}_{\text{kin}}\tilde{\mathbf{P}}, \qquad \mathbf{H}^{\tilde{\mathsf{Q}}} = E_{\text{exc}} + \alpha\tilde{\mathbf{Q}}\mathbf{H}_{\text{kin}}\tilde{\mathbf{Q}}. \tag{4}$$

In attractive systems[1] an expansion to first order is typically not sufficient as $\tilde{\mathbf{P}}\mathbf{H}_{\text{kin}}\tilde{\mathbf{P}}$ often vanishes: Intuitively, in a strongly attractive system the low-energy subspace $\tilde{\mathbf{P}}$ consists of tightly bound compound particles. If one applies the kinetic Hamiltonian $\mathbf{H}_{\text{kin}}$ to such a compound particle, it will move one particle outside the compound (thereby breaking it). The resulting state will be in the high-energy subspace $\tilde{\mathbf{Q}}$. Therefore, applying the projector $\tilde{\mathbf{P}}$ to $\mathbf{H}_{\text{kin}}\tilde{\mathbf{P}}$ will give 0. In order to obtain a non-trivial effective Hamiltonian in $\tilde{\mathsf{P}}$ one has to go to second order [47]:

$$\mathbf{H}^{\tilde{\mathsf{P}}} = -\frac{\alpha^2}{E_{\text{exc}}}\tilde{\mathbf{P}}\mathbf{H}_{\text{kin}}\tilde{\mathbf{Q}}\mathbf{H}_{\text{kin}}\tilde{\mathbf{P}}, \qquad \mathbf{H}^{\tilde{\mathsf{Q}}} = E_{\text{exc}} + \alpha\tilde{\mathbf{Q}}\mathbf{H}_{\text{kin}}\tilde{\mathbf{Q}}. \tag{5}$$

Note the emergent separation of time/energy-scales in the system: in the high-energy subspace (i.e. unbound particles) the typical time-scales for the evolution of the wavefunction is $1/\alpha$,

---

[1]The same argument can be made in repulsive systems, in which case the high-energy Hamiltonian $\mathbf{H}^{\tilde{\mathsf{Q}}}$ vanishes at first order.

while for the compound particles it is $1/\alpha^2$. Hamiltonian $\mathbf{H}^{\tilde{P}}$ in Eq. (5) can be interpreted as follows. In order to move a compound particle, one has to move all of its constituents. In case the compound particle consist of two particles, one first has to move one particle outside the pair and then in a second step has to move the second particle to rebuild the pair (therefore the Hamiltonian contains two times $\mathbf{H}_{\mathrm{kin}}$). Since the intermediate state is energetically not allowed, this process is heavily suppressed, which leads to a slow propagation of the pair. For example, in the Fermi-Hubbard chain Eq. (3) the effective hopping parameter of a pair $J_{\mathrm{pair}} = \frac{2J^2}{|U|} = 2\alpha J \ll J$ is much smaller than the single particle hopping $J$. This argument can also be extended to compound particles containing $N$ single particles, in which case we expect their first non-trivial Hamiltonian to be of order $\alpha^N$.

This already establishes the separation of scales between subssystems $\tilde{P}$ and $\tilde{Q}$. However, since $\tilde{P}$ and $\tilde{Q}$ have distinct energies, it is impossible to induce any scattering between both subsystems by introducing a (static) impurity into the system. This is not true anymore if the impurity is driven resonantly.

## 2.2 Ingredient 2: Floquet driving

Now we add an external Floquet-drive $\alpha\lambda\mathbf{V}(\omega t)$ to the system. The total Hamiltonian becomes

$$\mathbf{H} = \mathbf{H}_{\mathrm{int}} + \alpha\mathbf{H}_{\mathrm{kin}} + \alpha\lambda\mathbf{V}(\omega t). \tag{6}$$

Here $\mathbf{V}(\varphi)$ is a $2\pi$-periodic function and $\omega$ is the driving frequency. Note that the leading $\alpha$ in front of the driving term does not mean that the driving is small, we only scale it with $\alpha$ to ensure it has the same magnitude as the kinetic term. Similarly, we understand $\omega \sim \mathcal{O}(\alpha^0)$, meaning that $\omega$ is comparable to the energy difference between subspaces (which is required if we want to study resonant driving). For Floquet systems like Eq. (6) there exists a powerful framework, called Floquet theory, to study them. One important technique is the Floquet-Schrieffer-Wolff transformation [3,33], which extends the ideas of the last section to Floquet systems and allows us to find effective Hamiltonians for small $\alpha$. For our purposes the easiest way to introduce this transformation is to use the extended Hilbert space approach. The extended Hilbert space approach maps the driven system onto a static system. Unlike the high-frequency expansion, which involves an approximations, this map is exact, but comes with the price of introducing an extra dimension (in a nutshell, the extra dimension represents the Fourier component of the driven system).

The extended system can be described as follows: take the original system and create infinitely many copies of it, each copy labelled by an integer $a \in \mathbb{Z}$. Each of these copies gets an extra energy offset $-a\omega$. So far, the copies are uncoupled. We couple them using the Fourier components of the driving $\mathbf{V}(\omega t) = \sum_a \mathbf{V}_a e^{-ia\omega t}$: the coupling from copy $a$ to copy $b$ is given by $\mathbf{V}_{a-b}$. In short the interpretation of these couplings goes as follows: a jump from one copy $a$ to another copy $b$ describes the absorption of $b-a$ energy quanta of size $\omega$. One can therefore think of these extra layers as a neat way to keep track of the current energy of the system. A sketch of the extended system is given in Fig. 1 in the case of the driven Fermi-Hubbard chain with a driven impurity $\lambda\omega(\mathbf{n}_{0\uparrow} + \mathbf{n}_{0\downarrow})\cos\omega t$.

In case there is no Floquet-driving, the extended system consists of infinitely many independent subsystems $\tilde{P}$ and $\tilde{Q}$, one for each copy $a$. We denote them by $\tilde{P}_a$ and $\tilde{Q}_a$ respectively, with associated projectors $\tilde{\mathbf{P}}_a$ and $\tilde{\mathbf{Q}}_a$. Note that they have an energy offset $-a\omega$. If $E_{\mathrm{exc}} = A\omega$, where $A$ is an integer, then the driving is resonant, otherwise non-resonant.

Let us first discuss the non-resonant case: here all of the subsystems $P_a$ and $Q_a$ have well-separated energies in the extended Hilbert space. Therefore, one can apply the technique of the Schrieffer-Wolff transformation to the extended Hilbert space. The result for the effective Hamiltonians $\tilde{P}_0$ and $\tilde{Q}_0$ (the effective Hamiltonians in the other subspaces are the same up to

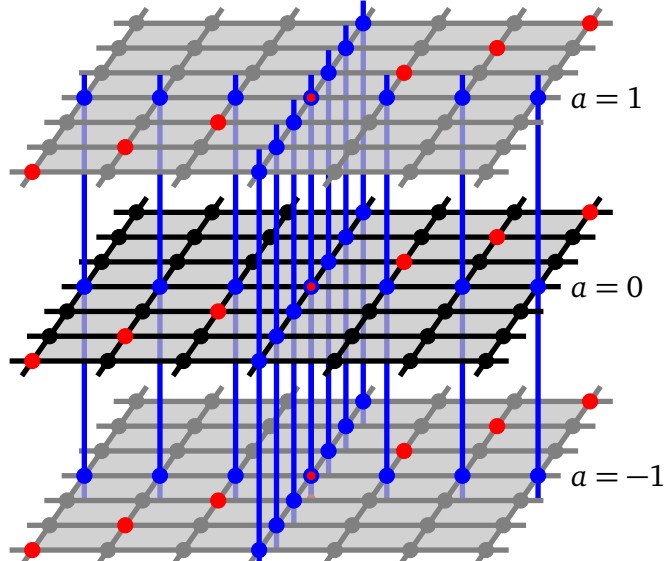

Figure 1: Sketch of the extended system for the two particle Fermi-Hubbard chain: the original system (black), though being a 1D system, is represented here by a 2D lattice $(n, m)$, where $n$ is the position of the first and $m$ the position of the second particle, with hopping $J$ between nearest neighbouring sites. The Hubbard interaction $U$ (red) acts on the diagonal. The extended system consists of infinite copies of this system (drawn in gray) each with an extra energy offset $-a\omega$. Neighbouring layers are coupled (blue) whenever one of the particles is at site 0 with coupling strength $\frac{\lambda\omega}{2}$, which represents a driving term of the form $\lambda\omega\big(\mathbf{n}_{0\uparrow} + \mathbf{n}_{0\downarrow}\big)\cos\omega t$.

the energy shift $-a\omega$) is the following:

$$\mathbf{H}^{\tilde{P}_0} = -\frac{\alpha^2}{E_{\text{exc}}} \tilde{\mathbf{P}}_0 \mathbf{H}_{\text{kin}} \tilde{\mathbf{Q}}_0 \mathbf{H}_{\text{kin}} \tilde{\mathbf{P}}_0 - \alpha^2\lambda^2 \sum_a \frac{\tilde{\mathbf{P}}_0 \mathbf{V}_{-a} \tilde{\mathbf{Q}}_a \mathbf{V}_a \tilde{\mathbf{P}}_0}{E_{\text{exc}} - a\omega}, \qquad \mathbf{H}^{\tilde{Q}_0} = E_{\text{exc}} + \alpha \tilde{\mathbf{Q}}_0 \mathbf{H}_{\text{kin}} \tilde{\mathbf{Q}}_0. \quad (7)$$

The first part of $\mathbf{H}^{\tilde{P}_0}$ is the same as Eq. (5), only the second part comes from the driving. The driving excites the system from $P_0$ to another copy $a$ and then in a second step back to $\tilde{P}_0$. In order to capture all possible excitations we need to sum over all copies $a$. Also note that $\mathbf{H}^{\tilde{Q}_0}$ is not affected to first order in $\alpha$. To conclude, in the non-resonant case, the physical situation is similar to the undriven case, only the precise expressions for the effective Hamiltonians are altered.

Let us now turn to the resonant driving case $E_{\text{exc}} = A\omega$ with integer $A$. First note that Eq. (7) cannot be applied in this case since it diverges. The problem is that both subsystems $\tilde{P}_0$ and $\tilde{Q}_A$ are both at the same energy in the extended Hilbert space. From the perspective of the driven model this means that an excitation from $\tilde{P}$ to $\tilde{Q}$ is no longer energetically forbidden, which is plausible as the system can now raise and lower its energy by any number of energy quanta $\omega$ due to the driving. Therefore, the subsystems $\tilde{P}_0$ and $\tilde{Q}_A$ can no longer be treated independently but instead they hybridize and their first order combined Hamiltonian is given by

$$\mathbf{H}^{\tilde{P}_0 \oplus \tilde{Q}_A} = \alpha \begin{pmatrix} 0 & \lambda \tilde{\mathbf{P}}_0 \mathbf{V}_{-A} \tilde{\mathbf{Q}}_A \\ \lambda \tilde{\mathbf{Q}}_0 \mathbf{V}_A \tilde{\mathbf{P}}_A & \tilde{\mathbf{Q}}_A \mathbf{H}_{\text{kin}} \tilde{\mathbf{Q}}_A \end{pmatrix}. \quad (8)$$

Note that the typical time-scale of this combined system is of order $1/\alpha$, due to the leading $\alpha$. Expression Eq. (8) has a very important physical consequence for bulk driving: due to

the driving compound particles may absorb energy from the driving and break apart. In fact, compound bound particles do not exist anymore in the system. We conclude that, in general, resonant bulk driving in strongly interacting systems will destroy the slow low-energy subspace.

## 2.3 Ingredient 3: scattering theory for resonantly driven impurities

This leads us to the peculiar case of a resonantly driven impurity: here the hybridization Eq. (8) only happens at the location of the impurity. Therefore, at the impurity the systems is described by Eq. (8), while away from the impurity it is described by Eq. (5).

On the boundary of the impurity there is a competition between the low-energy subspace trying to hybridize with the (fast) high-energy subspace and the intrinsic slow time-scale of the low-energy subspace. This competition makes the physical situation very interesting: imagine an incoming compound particle in the low-energy subspace. Due to its slow time evolution it will take a long time until it eventually reaches the impurity. Intuitively, as soon as the compound particle enters the impurity, there is a high change that it will absorb energy from the driving and break apart. Since the individual constituents evolve fast, they should propagate away to infinity before being able to recombine into a compound particle. Therefore, the naive expectation is that the impurity will break all incoming compound particles when reaching the impurity. In particular, in 1D systems where it is impossible to avoid scattering at the impurity, in the limit $\alpha \to 0$ all compound particles should be broken by the impurity. The surprising result of this paper is that this is not what is physically happening: instead, as $\alpha \to 0$, due to the separation of time-scales, the slow system on time-scales $1/\alpha^2$ and the fast system on time-scales $1/\alpha$ (which includes the hybridized impurity) still completely decouple. Therefore, the compound particle will not enter the impurity and break, but in fact will stay intact during scattering.

## 2.4 Summary of the physical situation

To summarize, we would like to study systems of the type

$$\mathbf{H}(t) = \alpha^2 \tilde{\mathbf{P}} \mathbf{h}^{\tilde{\mathbf{P}}} \tilde{\mathbf{P}} + A\omega \tilde{\mathbf{Q}} + \alpha \tilde{\mathbf{Q}} \mathbf{h}^{\tilde{\mathbf{Q}}} \tilde{\mathbf{Q}} + \alpha \mathbf{V}(\omega t), \tag{9}$$

where $\tilde{\mathbf{P}} \mathbf{h}^{\tilde{\mathbf{P}}} \tilde{\mathbf{P}}$ is the Hamiltonian of the low-energy band, $\tilde{\mathbf{Q}} \mathbf{h}^{\tilde{\mathbf{Q}}} \tilde{\mathbf{Q}}$ is the Hamiltonian of the high-energy band, $A\omega$ is the energy offset (band gap) between both bands and $\mathbf{V}(\omega t)$ is the resonantly driven impurity, which couples both bands. For the purpose of this paper it will be more natural to rescale time by a factor of $\alpha^2$ which gives a Hamiltonian of the form[2]

$$\mathbf{H}(t) = \tilde{\mathbf{P}} \mathbf{h}^{\tilde{\mathbf{P}}} \tilde{\mathbf{P}} + \frac{A\omega}{\alpha^2} \tilde{\mathbf{Q}} + \frac{1}{\alpha} \tilde{\mathbf{Q}} \mathbf{h}^{\tilde{\mathbf{Q}}} \tilde{\mathbf{Q}} + \frac{1}{\alpha} \mathbf{V}(\omega t / \alpha^2). \tag{10}$$

In order to understand these systems in the limit $\alpha \to 0$ we will first study static systems in the same regime:

$$\mathbf{H} = \mathbf{P} \mathbf{h}^{\mathbf{P}} \mathbf{P} + \frac{1}{\alpha} \mathbf{Q} \mathbf{h}^{\mathbf{Q}} \mathbf{Q} + \frac{1}{\alpha} \mathbf{V}, \tag{11}$$

where $\mathbf{V}$ is now a static impurity coupling a slow system P with a fast system Q (note that subsystems P and Q now have the same energy, so that scattering between them is allowed).[3] Later, in Section 5, we will extend the results to driven systems.

---

[2]Note that a rescaling of time does not affect the scattering at the impurity, as scattering theory is based on a long time limit.

[3]We denote subsystems for static impurities without tilde to distinguish them from those in driven systems.

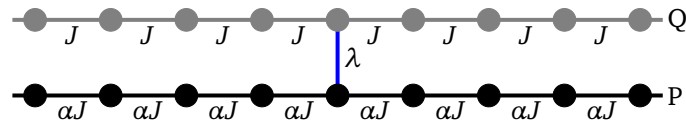

Figure 2: Sketch of the toy model: it consists of two chains, P and Q, with two different hopping amplitudes $\alpha J$ and $J$, where $\alpha \ll 1$. Both chains are coupled at site '0' with amplitude $\lambda$.

## 3  A simple toy model

To familiarize ourselves with the situation of scattering in systems with different time-scales let us start by studying a simple undriven toy model which shows the same separation of scales we expect from the driven case. The toy model is given by two infinite tight-binding chains, called P and Q. The hopping parameters on P and Q are given by $J_P = \alpha J$, $J_Q = J$, where $\alpha$ is a small positive real number. This means that a particle on P will propagate slower by a factor of $\alpha$ than a particle on Q (for instance, the typical group velocity of a particle in P is a factor $\alpha$ smaller than the group velocity in Q). Both chains are coupled at site '0', where a particle can hop from one chain to the other with amplitude $\lambda$. This resembles the effect of the resonant driving between a low-energy system P and a high-energy system Q (however, we want to stress again that this system is fully undriven). The full Hamiltonian of the toy model is given by

$$\mathbf{H}^{\text{toy}} = -\alpha J \mathbf{H}_P - J \mathbf{H}_Q + \lambda\big(\left|0_Q\right\rangle\left\langle 0_P\right| + \left|0_P\right\rangle\left\langle 0_Q\right|\big), \tag{12}$$

where $\left|n_{P/Q}\right\rangle$ denotes a particle in chain P/Q on site $n$ and

$$\mathbf{H}_{P/Q} = \sum_n \left|n_{P/Q}\right\rangle\left\langle n+1_{P/Q}\right| + \left|n+1_{P/Q}\right\rangle\left\langle n_{P/Q}\right| . \tag{13}$$

We also give a sketch of the situation in Fig. 2.

Our aim is to study the scattering of a particle initially in the slow chain P into the fast chain Q in the limit $\alpha \to 0$. In this limit the time-scales of both chains separate, i.e. the incoming particle in P will evolve slowly, but if it enters the other chain Q, it will evolve much faster there.

Note that the Hamiltonian Eq. (12) is difficult to interpret from the perspective of a particle in P, since as $\alpha \to 0$ the particle will become immobile. In the spirit of this paper it is natural to rescale $\mathbf{H}^{\text{toy}}$ by a factor $\frac{1}{\alpha}$ yielding:

$$\mathbf{H}^{\text{toy}} = -J \mathbf{H}_P - \frac{J}{\alpha} \mathbf{H}_Q + \frac{\lambda}{\alpha}\big(\left|0_Q\right\rangle\left\langle 0_P\right| + \left|0_P\right\rangle\left\langle 0_Q\right|\big). \tag{14}$$

Now as $\alpha \to 0$ a particle in P evolves normally, however it is coupled to a fast system Q via a strong coupling at site 0.

We want to stress that this rescaling of the Hamiltonian does not affect any of our results: if we were looking at real-time dynamics of this model the rescaling of energy would correspond to a rescaling of time. However, since we study scattering, which is based on a long time limit anyway, this rescaling is irrelevant. Indeed the Lippmann-Schwinger equations we are going to use later are completely invariant under rescaling of energy. From the viewpoint of scattering theory Eq. (12) and Eq. (14) are mathematically equivalent. The only advantage of Eq. (14) over Eq. (12) is that Eq. (14) describes the physical situation more intuitively, since it is expressed on the natural time-scale of an incoming particle in P.

## 3.1 Discussion of the naive expectation

Before we solve this problem exactly let us first think about the expected outcome as $\alpha \to 0$. Think of an incoming particle initially described by some wavepacket in the slow chain P. Because the dynamics on P are so slow it will only approach the scattering site '0' very slowly. If there was no link on site '0' the wavepacket would spend an increasingly long time at site '0'. But since there is a link, during its long time spend at site '0', some parts of the wavepacket will eventually hop to chain Q. Once they reach chain Q they will propagate much faster there and thus can quickly escape to infinity and will not come back to chain P. If $\alpha$ is sufficiently small this happens so fast that the still remaining part of the incoming wavepacket in the P chain basically has not moved at all. This process will repeat until the wavepacket in the P chain can finally leave site '0'. Since this time gets increasingly longer as $\alpha \to 0$ the P chain will be more and more depopulated. Thus, we expect that for small $\alpha$ that the majority of the wavepacket will end up in chain Q and only a minor fraction will remain in chain P. In the ultimate limit $\alpha \to 0$ we expect that the whole wavepacket will completely scatter into chain Q.

In the following we will solve the toy model exactly and show the naive expectation is wrong and that the situation is actually quite opposite: scattering from the slow chain P to the fast chain Q is suppressed for small $\alpha$ and completely vanishes for $\alpha \to 0$.

## 3.2 Solution of the Lippmann-Schwinger equations

Scattering in quantum mechanics can be studied using the Lippmann-Schwinger equation(s) [34]. They are equations about the scattering state $\left|\psi^+\right\rangle$, which is a superposition of an incoming wave and the corresponding outgoing waves. For the toy model the Lippmann-Schwinger equations are given by:

$$\mathbf{P}\left|\psi^+\right\rangle = |\phi\rangle + \frac{\lambda}{\alpha} \frac{1}{\varepsilon + J\mathbf{H}_\mathrm{P} + i\eta} |0_\mathrm{P}\rangle \left\langle 0_\mathrm{Q}\right| \mathbf{Q}\left|\psi^+\right\rangle, \tag{15}$$

$$\mathbf{Q}\left|\psi^+\right\rangle = \frac{\lambda/\alpha}{\varepsilon + \frac{J}{\alpha}\mathbf{H}_\mathrm{Q} + i\eta} |0_\mathrm{Q}\rangle \langle 0_\mathrm{P}| \mathbf{P}\left|\psi^+\right\rangle = \frac{\lambda}{\alpha\varepsilon + J\mathbf{H}_\mathrm{Q} + i\eta} |0_\mathrm{Q}\rangle \langle 0_\mathrm{P}| \mathbf{P}\left|\psi^+\right\rangle. \tag{16}$$

Here we introduced the projectors $\mathbf{P}$ and $\mathbf{Q}$ that project on the two chains P and Q, i.e.

$$\mathbf{P}|n_\mathrm{P}\rangle = |n_\mathrm{P}\rangle, \qquad\qquad \mathbf{P}\left|n_\mathrm{Q}\right\rangle = 0, \tag{17}$$

$$\mathbf{Q}|n_\mathrm{P}\rangle = 0, \qquad\qquad \mathbf{Q}\left|n_\mathrm{Q}\right\rangle = \left|n_\mathrm{Q}\right\rangle. \tag{18}$$

The state $|\phi\rangle$ is the incoming state given by a plane wave in chain P, i.e. $|\phi\rangle = \sum_n e^{ikn} |n_\mathrm{P}\rangle$ and $\frac{1}{E+J\mathbf{H}_\mathrm{P}+i\eta}$ and $\frac{1}{E+J\mathbf{H}_\mathrm{Q}+i\eta}$ denote the retarded Greens function of the uncoupled chains P and Q respectively (the infinitesimal $+i\eta$ is chosen such that the Greens function only produces outgoing states). Note that the energy of the incoming particle is given by $\varepsilon = -2J\cos k$.

To solve the Lippmann-Schwinger equations first insert Eq. (16) into Eq. (15) and obtain:

$$\mathbf{P}\left|\psi^+\right\rangle = |\phi\rangle + \frac{\lambda^2}{\alpha} \frac{1}{\varepsilon + J\mathbf{H}_\mathrm{P} + i\eta} |0_\mathrm{P}\rangle \left\langle 0_\mathrm{Q}\right| \frac{1}{\alpha\varepsilon + J\mathbf{H}_\mathrm{Q} + i\eta} \left|0_\mathrm{Q}\right\rangle \langle 0_\mathrm{P}| \mathbf{P}\left|\psi^+\right\rangle. \tag{19}$$

This equation can be thought of as an effective Lippmann-Schwinger equation for chain P alone. Apply $\langle 0_\mathrm{P}|$ on both sides:

$$\langle 0_\mathrm{P}| \mathbf{P}\left|\psi^+\right\rangle = \langle 0_\mathrm{P}|\phi\rangle + \frac{\lambda^2}{\alpha} \langle 0_\mathrm{P}| \frac{1}{\varepsilon + J\mathbf{H}_\mathrm{P} + i\eta} |0_\mathrm{P}\rangle \left\langle 0_\mathrm{Q}\right| \frac{1}{\alpha\varepsilon + J\mathbf{H}_\mathrm{Q} + i\eta} \left|0_\mathrm{Q}\right\rangle \langle 0_\mathrm{P}| \mathbf{P}\left|\psi^+\right\rangle, \tag{20}$$

and solve for $\langle 0_\mathrm{P}| \mathbf{P}\left|\psi^+\right\rangle$:

$$\langle 0_\mathrm{P}| \mathbf{P}\left|\psi^+\right\rangle = \frac{1}{1 - \frac{\lambda^2}{\alpha} \langle 0_\mathrm{P}| \frac{1}{\varepsilon + J\mathbf{H}_\mathrm{P} + i\eta} |0_\mathrm{P}\rangle \left\langle 0_\mathrm{Q}\right| \frac{1}{\alpha\varepsilon + J\mathbf{H}_\mathrm{Q} + i\eta} \left|0_\mathrm{Q}\right\rangle} \langle 0_\mathrm{P}|\phi\rangle. \tag{21}$$

Plugging the last result into Eq. (19) and Eq. (16) gives the full wavefunction $\left|\psi^+\right\rangle$ for any finite value of $\alpha$. In the limit we are interested in, $\alpha \to 0$, we find the following:

$$\langle 0_P| \mathbf{P} \left|\psi^+\right\rangle = -\frac{\alpha}{\lambda^2 \langle 0_P| \frac{1}{\varepsilon + J\mathbf{H}_P + i\eta} |0_P\rangle \langle 0_Q| \frac{1}{J\mathbf{H}_Q + i\eta} |0_Q\rangle} \langle 0_P|\phi\rangle + \mathcal{O}(\alpha^2) \to 0, \qquad (22)$$

and by using Eq. (19) and Eq. (16):

$$\mathbf{P}\left|\psi^+\right\rangle = |\phi\rangle - \frac{1}{\varepsilon + J\mathbf{H}_P + i\eta} |0_P\rangle \frac{1}{\langle 0_P| \frac{1}{\varepsilon + J\mathbf{H}_P + i\eta} |0_P\rangle} \langle 0_P|\phi\rangle + \mathcal{O}(\alpha), \qquad (23)$$

$$\mathbf{Q}\left|\psi^+\right\rangle = -\alpha \frac{\lambda}{J\mathbf{H}_Q + i\eta} |0_Q\rangle \frac{1}{\lambda^2 \langle 0_P| \frac{1}{\varepsilon + J\mathbf{H}_P + i\eta} |0_P\rangle \langle 0_Q| \frac{1}{J\mathbf{H}_Q + i\eta} |0_Q\rangle} \langle 0_P|\phi\rangle + \mathcal{O}(\alpha^2) \to 0. \quad (24)$$

The last result Eq. (24) is quite surprising as it shows that $\mathbf{Q}\left|\psi^+\right\rangle$ vanishes in the limit $\alpha \to 0$, completely contrary to our naive expectation.

What is the reason for the suppression of scattering into Q? It is an implication of the observation that the wavefunction $\mathbf{P}\left|\psi^+\right\rangle$ vanishes at $|0_P\rangle$. We will also encounter a similar observation in the general case and we will call it 'artificial boundary condition' since sending $\langle 0_P| \mathbf{P} \left|\psi^+\right\rangle \to 0$ is the same as imposing the boundary condition that the wavefunction has to vanish on $\langle 0_P| \mathbf{P} \left|\psi^+\right\rangle$ throughout the scattering. With this picture in mind we can easily understand why scattering into the Q chain cannot occur: in order for the particle to jump into chain Q it first has to reach site $|0_P\rangle$. But since the wavefunction is forced to zero at $|0_P\rangle$ this cannot happen and thus there is no scattering into the Q chain.

There is also a third interesting observation: bBecause the particle cannot enter site $|0_P\rangle$ it can also not reach the other side of chain P, but is necessarily always reflected. This means the transmission vanishes for all incoming particles independent of their initial momentum k.

Before we go on to the general case let us make some remarks:

- Note that the result is independent of the coupling strength $\lambda$, in particular it holds for arbitrarily small $\lambda$. Of course if $\lambda = 0$ then there is no scattering at all. This shows that the two limits $\lambda \to 0$ and $\alpha \to 0$ are not interchangeable. In practice this means that our result only applies if $\alpha \ll \lambda$. In this sense our result is a non-perturbative result in the coupling strength $\lambda$. Indeed it is not possible to obtain the same result by first doing a perturbative expansion in $\lambda$ and then sending $\alpha \to 0$ (already the second order correction is infinite).

- Similarly, our result only applies if the energy of the incoming particle is not too close to the boundaries of the band $\varepsilon = \pm 2J$. The Greens operator of a tight-binding chain can be evaluated explicitly (see for example [48] (9.93) - (9.98)):

$$\langle n_P| \frac{1}{\varepsilon + J\mathbf{H}_P + i\eta} |m_P\rangle = \frac{e^{i|k||n-m|}}{2i \sin|k|}, \qquad \varepsilon = -2J \cos k. \qquad (25)$$

This expression is finite inside the band, but diverges for $\varepsilon \to \pm 2J$. In fact, for the toy model $\varepsilon = \pm 2J$ still gives Eq. (23) and Eq. (24), but this is not clear in the general case. For the interested reader, we give the explicit expressions for the transmission probabilities for the toy model in Appendix A and plot them for different values of $\alpha$.

- Instead of studying an incoming particle in P one could also study an incoming particle in Q. In this case the more natural Hamiltonian is Eq. (12) since it is expressed on Q's natural timescale. As $\alpha \to 0$ chain P is completely trivial and decouples from the system. Due to the impurity, chain Q is still coupled to the single state $|0\rangle_P$ with energy 0. Situations like that are well studied and lead to a Fano-resonance (transmission vanishes for

an incoming particle with energy 0) [49, 50]. We discuss the relationship between our result and the Fano-resonance in Appendix B. Note that a Fano-resonance in Q appears for the toy model but does not necessarily occur in the general case.

In the following we will discuss the situation of scattering from a slow part of a system into a fast part in a general system. We will find similar results as in the toy model. We will again find 'artificial boundary conditions', namely that certain parts of the wavefunction in the slow part are going to vanish as $\alpha \to 0$, which will in turn imply that scattering into the fast system is suppressed. Interestingly the third observation, namely that all incoming particles are reflected, is not true in general. This will only happen when the artificial boundary conditions are such that they disconnect subsystem P.

## 4  General result for static impurities

In this section we now extend the results for the toy model to a general static setting: consider a system with two subsystems: a slow subsystem P and a fast subsystem Q, described by projectors $\mathbf{P}$ and $\mathbf{Q}$. Slow and fast means that there is a small parameter $\alpha$, such that the uncoupled Hamiltonians on P and Q are given by $\mathbf{h}_\alpha^P$ and $\frac{1}{\alpha}\mathbf{h}_\alpha^Q$, where $\mathbf{h}_\alpha^{P/Q}$ are supposed to stay finite as $\alpha \to 0$ (in other words P consists of narrow bands and Q consists of broad bands). Furthermore there is a (possibly $\alpha$ dependent) potential $\frac{1}{\alpha}\mathbf{V}_\alpha$ which couples both subsystems ($\mathbf{V}_\alpha$ is also supposed to stay finite as $\alpha \to 0$). The $1/\alpha$ in front of the potential means that the coupling potential is on the same time-scale as the fast subsystem Q. The full Hamiltonian can either be expressed on the time-scale of the slow subsystem P:

$$\mathbf{H}_\alpha = \mathbf{h}_\alpha^P + \tfrac{1}{\alpha}\mathbf{h}_\alpha^Q + \tfrac{1}{\alpha}\mathbf{V}_\alpha \,, \tag{26}$$

where it describes a particle strongly coupled to a very fast system Q, or equivalently on the time-scale of the fast subsystem Q:

$$\mathbf{H}_\alpha = \alpha\mathbf{h}_\alpha^P + \mathbf{h}_\alpha^Q + \mathbf{V}_\alpha \,, \tag{27}$$

where it describes a slow particle in P coupled to a subsystem Q. Again, from the viewpoint of scattering theory both Hamiltonians are equivalent and are described by the same Lippmann-Schwinger equations. For the purpose of this paper we prefer Eq. (26) since it describes the physical situation better for a particle in P: from the perspective of a particle in P, subsystem P is strongly coupled to a fast subsystem/high-energy subsystem Q.

The incoming particle is initially in the slow subsystem P and is described by an incoming eigenstate $|\phi\rangle$ of $\mathbf{h}_\alpha^P$ with eigenenergy $\varepsilon_\alpha$ (Note that we will for simplicity assume that $|\phi\rangle$ is an eigenstate of $\mathbf{h}_\alpha^P$ for all $\alpha$ – if this is not the case one can easily achieve it by first applying an $\alpha$ dependent unitary transformation which diagonalizes $\mathbf{h}_\alpha^P$). The Lippmann-Schwinger equations for this situation look as follows:

$$\mathbf{P}\,|\psi_\alpha^+\rangle = |\phi\rangle + \frac{1}{\alpha}\frac{1}{\varepsilon_\alpha - \mathbf{h}_\alpha^P + i\eta}\big(\mathbf{P}\mathbf{V}_\alpha\mathbf{P}\,|\psi_\alpha^+\rangle + \mathbf{P}\mathbf{V}_\alpha\mathbf{Q}\,|\psi_\alpha^+\rangle\big)\,, \tag{28}$$

$$\mathbf{Q}\,|\psi_\alpha^+\rangle = \frac{1}{\varepsilon_\alpha - \frac{1}{\alpha}\mathbf{h}_\alpha^Q + i\eta}\frac{1}{\alpha}\big(\mathbf{Q}\mathbf{V}_\alpha\mathbf{P}\,|\psi_\alpha^+\rangle + \mathbf{Q}\mathbf{V}_\alpha\mathbf{Q}\,|\psi_\alpha^+\rangle\big)$$

$$= \frac{1}{\alpha\varepsilon_\alpha - \mathbf{h}_\alpha^Q + i\eta}\big(\mathbf{Q}\mathbf{V}_\alpha\mathbf{P}\,|\psi_\alpha^+\rangle + \mathbf{Q}\mathbf{V}_\alpha\mathbf{Q}\,|\psi_\alpha^+\rangle\big)\,. \tag{29}$$

We can solve the second equation Eq. (29) by introducing the Greens operator $\mathbf{G}_\alpha^Q$ of subsystem Q:

$$\mathbf{Q}\,|\psi_\alpha^+\rangle \overset{\text{def}}{=} \mathbf{Q}\mathbf{G}_\alpha^Q\mathbf{Q}\mathbf{V}_\alpha\mathbf{P}\,|\psi_\alpha^+\rangle\,. \tag{30}$$

The interpretation of $\mathbf{G}_\alpha^Q$ is as follows: imagine we would for some reason know the full wavefunction in subsystem P. Then we could compute the wavefunction in subsytem Q using Eq. (30). Formally the Greens operator is given by

$$\mathbf{G}_\alpha^Q = \mathbf{Q} \frac{1}{\alpha \varepsilon_\alpha - \mathbf{h}_\alpha^Q - \mathbf{Q}\mathbf{V}_\alpha\mathbf{Q} + i\eta} \mathbf{Q}, \tag{31}$$

but we do not need its explicit form. By inserting Eq. (30) into Eq. (28) and denoting $\mathbf{G}_\alpha^P = \frac{1}{\varepsilon_\alpha - \mathbf{h}_\alpha^P + i\eta}$ we find

$$\mathbf{P}\left|\psi_\alpha^+\right\rangle = |\phi\rangle + \frac{1}{\alpha}\mathbf{G}_\alpha^P\mathbf{V}_\alpha^{\text{eff}}\mathbf{P}\left|\psi_\alpha^+\right\rangle, \tag{32}$$

which is an effective Lippmann-Schwinger equation for subsystem P only, with an effective potential

$$\mathbf{V}_\alpha^{\text{eff}} = \mathbf{P}\mathbf{V}_\alpha\mathbf{P} + \mathbf{P}\mathbf{V}_\alpha\mathbf{Q}\mathbf{G}_\alpha^Q\mathbf{Q}\mathbf{V}_\alpha\mathbf{P}, \tag{33}$$

that also includes an induced effective potential from subsystem Q (in particle physics the effective potential is often called the optical potential, see e.g. [51]). Note that the effective potential $\mathbf{V}_\alpha^{\text{eff}}$ is not necessarily a hermitian operator (because the Greens operator might not be hermitian either). The non-hermitian part of $\mathbf{V}_\alpha^{\text{eff}}$ describes particle loss from subsystem P to subsystem Q, i.e. the part of the wavefunction that scatters from P into Q and then propagates away to infinity.

Recall that we would like to study the limit $\alpha \to 0$. We can expect that the effective potential has a finite limit $\mathbf{V}_0^{\text{eff}}$, because both $\mathbf{V}_\alpha$ and $\mathbf{G}_\alpha^Q$ should have a finite limit too. We will analyse the limit of the Lippmann-Schwinger equations in two steps: first, we will show a theorem about the effective Lippmann-Schwinger equation Eq. (32) and then apply that result in a second theorem to show that the wavefunction in subsystem Q goes to zero as $\alpha \to 0$.

We will not need many assumptions about the appearing quantities. First, we will naturally require that the problem is such that it can be described by the Lippmann-Schwinger equation. Second we need that all appearing quantities either have a finite limit or a Taylor expansion up to first in $\alpha$. Third we also need some additional technical assumptions (a)-(e). They will become clear during the proofs and state that certain operators are invertible. We state them in Appendix E and also motivate why we expect them to hold in a generic system.

Before we state the theorems we want to introduce the following terminology: the space on which a linear operator $\mathbf{A}$ acts is the orthogonal of the kernel of $\mathbf{A}$, i.e. intuitively the subspace on which $\mathbf{A}$ 'does not vanish' (as opposed to the kernel which is the space on which $\mathbf{A}$ vanishes).

We will now define the most important projector of this paper: we will call $\mathbf{K}$ the projector on the space where $\mathbf{V}_0^{\text{eff}}$ acts. Note that it is space is a subspace of P, and thus $\mathbf{KP} = \mathbf{PK} = \mathbf{K}$. Furthermore, we will need the following two projectors as well: $\bar{\mathbf{K}} = \mathbf{P} - \mathbf{K}$ as the projector on the kernel of $\mathbf{V}_0^{\text{eff}}$ and $\mathbf{L}$ as the projector on the space on which $\bar{\mathbf{K}}\mathbf{V}_0^{\text{eff}'}\bar{\mathbf{K}}$ acts.

We can now go on to state the first main result of this paper. Note that conclusion (i) is the most important one, conclusions (ii) and (iii) merely help to do actual computations.

**Theorem 1** *Consider a scattering problem with uncoupled Hamiltonian $\mathbf{h}_\alpha^P$ and (not necessarily hermitian) potential $\frac{1}{\alpha}\mathbf{V}_\alpha^{\text{eff}}$. The incoming state $|\phi\rangle$ is an eigenstate of $\mathbf{h}_\alpha^P$ with eigenvalue $\varepsilon_\alpha$. Assume that $\mathbf{h}_\alpha^P$ has a finite limit for $\alpha \to 0$ and that $\mathbf{V}_\alpha^{\text{eff}} = \mathbf{V}_0^{\text{eff}} + \alpha\mathbf{V}_0^{\text{eff}'} + \mathcal{O}(\alpha^2)$ has a Taylor expansion up to first order.*

*In case that $\mathbf{V}_0^{\mathrm{eff}} \neq 0$ and the technical assumptions (a), (b) hold we find in the limit $\alpha \to 0$:*

*(i)* $\mathbf{K}\left|\psi_\alpha^+\right\rangle \to 0$.

*Furthermore, we have the following explicit formulas for the limiting wavefunction:*

*(ii) If additionally $\bar{\mathbf{K}}\mathbf{V}_0^{\mathrm{eff}} = 0$ and (d), (e) hold:*

$$\mathbf{L}\left|\psi_0^+\right\rangle = \frac{1}{\mathbf{L} - \left(\mathbf{L} - \mathbf{L}\mathbf{G}_0^{\mathrm{P}}\mathbf{K}\frac{1}{\mathbf{K}\mathbf{G}_0^{\mathrm{P}}\mathbf{K}}\mathbf{K}\right)\mathbf{K}\mathbf{G}_0^{\mathrm{P}}\bar{\mathbf{K}}\mathbf{V}_0^{\mathrm{eff}'}\mathbf{L}}\left(\mathbf{L} - \mathbf{L}\mathbf{G}_0^{\mathrm{P}}\mathbf{K}\frac{1}{\mathbf{K}\mathbf{G}_0^{\mathrm{P}}\mathbf{K}}\mathbf{K}\right)|\phi\rangle . \qquad (34)$$

*(iii) If additionally $\bar{\mathbf{K}}\mathbf{V}_0^{\mathrm{eff}} = 0$ and (d) holds:*

$$\mathbf{P}\left|\psi_0^+\right\rangle = \left(1 - \mathbf{G}_0^{\mathrm{P}}\mathbf{K}\frac{1}{\mathbf{K}\mathbf{G}_0^{\mathrm{P}}\mathbf{K}}\mathbf{K}\right)\left(|\phi\rangle + \mathbf{G}_0^{\mathrm{P}}\bar{\mathbf{K}}\mathbf{V}_0^{\mathrm{eff}'}\mathbf{L}\left|\psi_0^+\right\rangle\right). \qquad (35)$$

Note that $\mathbf{L}$ can be empty (as it is for example for the toy model) and then conclusion (ii) is irrelevant and conclusion (iii) simplifies to $\mathbf{P}\left|\psi_0^+\right\rangle = \left(1 - \mathbf{G}_0^{\mathrm{P}}\mathbf{K}\frac{1}{\mathbf{K}\mathbf{G}_0^{\mathrm{P}}\mathbf{K}}\mathbf{K}\right)|\phi\rangle$.

The proof of Theorem 1 is given in Appendix C. Its most important conclusion is the first one. It says that certain parts of the wavefunction, namely exactly those where the effective potential acts, will vanish in the limit. This is the 'artificial boundary condition' which we already encountered in the toy model.

The other two conclusions are only relevant in case one would like to evaluate the full wavefunction in the limit $\alpha \to 0$ explicitly (for example in order to obtain the transmission amplitude). Even though they might seem complicated expression they are actually helpful because for localized scattering potentials, $\mathbf{K}$ and $\mathbf{L}$ project onto finite dimensional subspaces. Therefore, all quantities appearing in conclusion (ii) are either finite dimensional vectors or matrices which can be handled much more easily than infinite dimensional operators. Note that the extra requirement $\bar{\mathbf{K}}\mathbf{V}_0^{\mathrm{eff}} = 0$ for conclusion (ii) and (iii) is automatically satisfied if $\mathbf{V}_0^{\mathrm{eff}}$ is hermitian.

Now we are ready to study the implications of Theorem 1 in subsystem Q. The idea is, as in the toy model, that the wavefunction in subsystem Q will vanish because the potential connecting P and Q only depends on the wavefunction in a small part $\mathbf{K}\left|\psi_\alpha^+\right\rangle$ and because Theorem 1 already shows us that the wavefunction vanishes on exactly that part $\mathbf{K}\left|\psi_\alpha^+\right\rangle \to 0$. In order to see this we need to first study the relation between $\mathbf{V}_\alpha^{\mathrm{eff}}$ and $\mathbf{Q}\mathbf{V}_\alpha\mathbf{P}$. To this end define $\mathbf{K}^{\mathrm{P}}$ as the projector onto the space on which $\mathbf{Q}\mathbf{V}_0\mathbf{P}$ acts and $\mathbf{K}^{\mathrm{Q}}$ as the projector onto the image of $\mathbf{Q}\mathbf{V}_0\mathbf{P}$. This means that $\mathbf{Q}\mathbf{V}_0\mathbf{P} = \mathbf{K}^{\mathrm{Q}}\mathbf{V}_0\mathbf{K}^{\mathrm{P}}$. The following lemma provides some useful relations in the case $\mathbf{P}\mathbf{V}_\alpha\mathbf{P} \to 0$.

**Lemma 1** *Assume that $\mathbf{G}_\alpha^{\mathrm{Q}} = \mathbf{G}_0^{\mathrm{Q}} + \alpha\mathbf{G}_0^{\mathrm{Q}'} + \mathcal{O}(\alpha^2)$ and $\mathbf{V}_\alpha = \mathbf{V}_0 + \alpha\mathbf{V}_0' + \mathcal{O}(\alpha^2)$ have a Taylor expansion at least up to first order. In case that $\mathbf{G}_0^{\mathrm{Q}} \neq 0$, $\mathbf{P}\mathbf{V}_0\mathbf{P} = 0$, $\mathbf{Q}\mathbf{V}_0\mathbf{P} \neq 0$ and that technical assumption (c) holds, we find in the limit $\alpha \to 0$:*

- *$\mathbf{V}_\alpha^{\mathrm{eff}}$ has a Taylor expansion up to first order $\mathbf{V}_\alpha^{\mathrm{eff}} = \mathbf{V}_0^{\mathrm{eff}} + \alpha\mathbf{V}_0^{\mathrm{eff}'} + \mathcal{O}(\alpha^2)$,*

- *$\mathbf{V}_0^{\mathrm{eff}} \neq 0$,*

- *$\mathbf{K} = \mathbf{K}^{\mathrm{P}}$,*

- *$\bar{\mathbf{K}}\mathbf{V}_0^{\mathrm{eff}} = 0$,*

- *$\bar{\mathbf{K}}\mathbf{V}_0^{\mathrm{eff}'}\bar{\mathbf{K}} = \bar{\mathbf{K}}\mathbf{V}_0'\bar{\mathbf{K}} = \mathbf{L}\mathbf{V}_0'\mathbf{L}$.*

The proof is done in Appendix D. Note that $\bar{\mathbf{K}}\mathbf{V}_0^{\text{eff}} = 0$ is exactly the requirement for conclusions (ii) and (iii) of Theorem 1. Using Lemma 1 it is easy to obtain the following result:

**Theorem 2** *Consider a scattering problem with uncoupled Hamiltonian* $\mathbf{H}_0 = \mathbf{P}\mathbf{h}_\alpha^{\text{P}}\mathbf{P} + \frac{1}{\alpha}\mathbf{Q}\mathbf{h}_\alpha^{\text{Q}}\mathbf{Q}$, *where* $\mathbf{P}$ *and* $\mathbf{Q}$ *are complementary projectors* $\mathbf{P} + \mathbf{Q} = 1$, *and (hermitian) potential* $\frac{1}{\alpha}\mathbf{V}_\alpha$. *The incoming state* $|\phi\rangle$ *is an eigenstate of* $\mathbf{h}_\alpha^{\text{P}}$ *with eigenvalue* $\varepsilon_\alpha$. *Assume that* $\mathbf{h}_\alpha^{\text{P}}$ *has a finite limit for* $\alpha \to 0$ *and that* $\mathbf{G}_\alpha^{\text{Q}} = \mathbf{G}_0^{\text{Q}} + \alpha\mathbf{G}_0^{\text{Q}'} + \mathcal{O}(\alpha^2)$ *and* $\mathbf{V}_\alpha = \mathbf{V}_0 + \alpha\mathbf{V}_0' + \mathcal{O}(\alpha^2)$ *have a Taylor expansion at least up to first order.*

*In case that* $\mathbf{G}_0^{\text{Q}} \neq 0$, $\mathbf{P}\mathbf{V}_0\mathbf{P} = 0$, $\mathbf{Q}\mathbf{V}_0\mathbf{P} \neq 0$ *and the technical assumptions (a), (b) and (c) hold we find in the limit* $\alpha \to 0$:

(i) $\mathbf{K}|\psi_\alpha^+\rangle \to 0$,

(iv) $\mathbf{Q}|\psi_\alpha^+\rangle \to 0$.

*Furthermore, we have similar formulas as in Theorem 1:*

(ii) *If additionally (d) and (e) hold:*

$$\mathbf{L}|\psi_0^+\rangle = \frac{1}{\mathbf{L} - \left(\mathbf{L} - \mathbf{L}\mathbf{G}_0^{\text{P}}\mathbf{K}\frac{1}{\mathbf{K}\mathbf{G}_0^{\text{P}}\mathbf{K}}\mathbf{K}\right)\mathbf{K}\mathbf{G}_0^{\text{P}}\mathbf{L}\mathbf{V}_0'\mathbf{L}}\left(\mathbf{L} - \mathbf{L}\mathbf{G}_0^{\text{P}}\mathbf{K}\frac{1}{\mathbf{K}\mathbf{G}_0^{\text{P}}\mathbf{K}}\mathbf{K}\right)|\phi\rangle. \tag{36}$$

(iii) *If additionally (d) holds:*

$$\mathbf{P}|\psi_0^+\rangle = \left(1 - \mathbf{G}_0^{\text{P}}\mathbf{K}\frac{1}{\mathbf{K}\mathbf{G}_0^{\text{P}}\mathbf{K}}\mathbf{K}\right)\left(|\phi\rangle + \mathbf{G}_0^{\text{P}}\mathbf{L}\mathbf{V}_0'\mathbf{L}|\psi_0^+\rangle\right). \tag{37}$$

Proof: Due to Lemma 1 we can apply Theorem 1 which directly gives conclusions (i), (ii) and (iii). Using $\mathbf{K}|\psi_\alpha^+\rangle \to 0$, the definition of the Greens operator Eq. (30) and $\mathbf{K}^{\text{P}} = \mathbf{K}$ we find that the wavefunction in subsystem Q is given by:

$$\mathbf{Q}|\psi_\alpha^+\rangle = \mathbf{G}_\alpha^{\text{Q}}\mathbf{Q}\mathbf{V}_\alpha\mathbf{P}|\psi_\alpha^+\rangle \to \mathbf{G}_0^{\text{Q}}\mathbf{K}^{\text{Q}}\mathbf{V}_0\mathbf{K}^{\text{P}}|\psi_0^+\rangle = \mathbf{G}_0^{\text{Q}}\mathbf{K}^{\text{Q}}\mathbf{V}_0\mathbf{K}|\psi_0^+\rangle = 0. \tag{38}$$

This concludes the theorem. Result (iv) shows that the wavefunction completely vanishes in subsystem Q, which intermediately implies that the surprising suppression of scattering from the slow subspace P into the fast subspace Q in the limit of $\alpha \to 0$ is a rather general feature.

## 4.1 Discussion of the requirements

There are two important requirements for the theorems to hold: first, we of course need that the uncoupled system consists of two subsystems that evolve on different time-scales. Second we require that $\mathbf{V}_\alpha^{\text{eff}}$ does not vanish as $\alpha \to 0$. This basically says that the effective potential and thus also the bare potential $\mathbf{V}_\alpha$ has the same energy/timescale as subsystem Q. This scaling is crucial. For example we could also consider the case when the effective potential is on the same scale as subsystem P, i.e. $\mathbf{V}_\alpha^{\text{eff}} \sim \alpha$. This can happen for instance if the bare potential has an intermediate scale $\mathbf{V}_\alpha \sim \sqrt{\alpha}$. Then we could directly take the limit in Eq. (32) and obtain $\mathbf{P}|\psi_0^+\rangle = |\phi\rangle + \mathbf{G}_0^{\text{P}}\mathbf{V}_0^{\text{eff}'}\mathbf{P}|\psi_0^+\rangle$ which is just the Lippmann-Schwinger equation for the effective Hamiltonian $\mathbf{h}_0^{\text{P}} + \mathbf{V}_0^{\text{eff}'}$. If this effective Hamiltonian is hermitian it describes unitary dynamics in subsystem P and thus scattering into Q vanishes. If on the other hand the effective Hamiltonian contains a non-hermitian part then there will be non-vanishing scattering from P to Q.

A further requirement of our results are that all appearing quantities, in particular the Greens operators, are finite (i.e. they do not have divergences). As we observed in the discussion of the toy model Eq. (25), Greens operators can diverge at certain energies, for instance close to the boundaries of an energy band. In this case our result is not applicable. However, by splitting the Greens operator into a divergent and a finite part, it should still be possible to obtain analytical results by adapting the steps in the proof of Theorem 1 and treating the divergent part separately.

Apart from the requirements, our result is completely general. For instance, it does not only apply to single-particle systems, but also to many-particle systems. In fact, in Appendix G we will discuss a two-particle system.

## 5  Extension to resonantly Floquet-driven impurities

In this section we will now extend the results for static impurities to the case of resonantly driven impurities. While the situation for static impurities is quite unusual, due to the fact that two bands with significantly different bandwidths need to overlap (in order to allow for scattering), such situations are easily constructed for driven impurities: choose a system with two bands, whose bandwidths differ and couple them using a resonantly driven impurity. This will effectively make both bands to overlap and therefore scattering between them is allowed. Denote by $\alpha$ the ratio of the two bandwidths and by $\omega(\alpha)$ the driving frequency, which is allowed to depend on $\alpha$. The results of this paper will apply in the limit of both $\alpha \to 0$ and high-frequency driving $\omega(\alpha) \gtrsim 1/\alpha^2$. For later reference let us define

$$\frac{1}{\omega(\alpha)} = \frac{\alpha^2}{\omega_0} + \mathcal{O}(\alpha^3), \tag{39}$$

where we also allow $\omega_0$ to be infinite $\omega_0 = \infty$ (in which case the driving frequency is even higher $\omega(\alpha) \gg 1/\alpha^2$).

To allow our results to cover very general systems, we will allow for the presence of more bands. In general, we will separate the system into two subsystems $\tilde{P}$ and $\tilde{Q}$ (with associated projectors $\tilde{\mathbf{P}}$ and $\tilde{\mathbf{Q}}$), both of which can contain multiple energy bands. Overall the dynamics in $\tilde{P}$ have to be much slower than in $\tilde{Q}$ (meaning that a typical bandwidth in $\tilde{P}$ is much smaller than a typical bandwidth in $\tilde{Q}$). Denote the ratio of both time-scales as $\alpha = \frac{T_{\text{fast}}}{T_{\text{slow}}} \ll 1$.

The total Hamiltonian of the system including impurity is

$$\mathbf{H}(t) = \tilde{\mathbf{P}}\mathbf{h}_\alpha^{\tilde{P}}\tilde{\mathbf{P}} + A\omega(\alpha)\tilde{\mathbf{Q}} + \frac{1}{\alpha}\tilde{\mathbf{Q}}\mathbf{h}_\alpha^{\tilde{Q}}\tilde{\mathbf{Q}} + \frac{1}{\alpha}\mathbf{V}_\alpha(\omega(\alpha)t). \tag{40}$$

This is a generalization of Eq. (10): here $\tilde{\mathbf{P}}\mathbf{h}_\alpha^{\tilde{P}}\tilde{\mathbf{P}}$ is the Hamiltonian of subsystem $\tilde{P}$, $\frac{1}{\alpha}\tilde{\mathbf{Q}}\mathbf{h}_\alpha^{\tilde{Q}}\tilde{\mathbf{Q}}$ is the Hamiltonian of subsystem $\tilde{Q}$ and $A\omega(\alpha)$ is the energy offset between both systems ($A \in \mathbb{Z}$). Both subsystems are coupled by a driven impurity $\frac{1}{\alpha}\mathbf{V}_\alpha(\omega(\alpha)t)$ which by construction is at resonance with the energy difference between $\tilde{P}$ and $\tilde{Q}$. We assume that the static part of the driven impurity does not affect $\tilde{P}$, i.e. $\tilde{\mathbf{P}}\mathbf{V}_{0;0}\tilde{\mathbf{P}} = \tilde{\mathbf{P}}\left[\int_0^{2\pi}\frac{\mathrm{d}\varphi}{2\pi}\mathbf{V}_0(\varphi)\right]\tilde{\mathbf{P}} = 0$. As an example for Hamiltonians of the form Eq. (40), consider Hamiltonians constructed via the Schrieffer-Wolff transformation as in Section 2.

In the extended Hilbert space the Lippmann-Schwinger equations associated with this system can be written as follows:

$$\tilde{\mathbf{P}}_a \left|\psi_\alpha^+\right\rangle = \left|\phi\right\rangle \delta_{a,0} + \frac{1}{\varepsilon_\alpha - \mathbf{h}_\alpha^{\tilde{\mathrm{P}}} + a\omega(\alpha) + i\eta} \frac{1}{\alpha}\left[\sum_b \tilde{\mathbf{P}}_a \mathbf{V}_{\alpha;a\leftarrow b}\tilde{\mathbf{P}}_b \left|\psi_\alpha^+\right\rangle + \tilde{\mathbf{P}}_a \mathbf{V}_{\alpha;a\leftarrow b}\tilde{\mathbf{Q}}_b \left|\psi_\alpha^+\right\rangle\right],$$
(41)

$$\tilde{\mathbf{Q}}_a \left|\psi_\alpha^+\right\rangle = \frac{1}{\varepsilon_\alpha - \frac{1}{\alpha}\mathbf{h}_\alpha^{\tilde{\mathrm{Q}}} + (a-A)\omega(\alpha) + i\eta} \frac{1}{\alpha}\left[\sum_b \tilde{\mathbf{Q}}_a \mathbf{V}_{\alpha;a\leftarrow b}\tilde{\mathbf{P}}_b \left|\psi_\alpha^+\right\rangle + \tilde{\mathbf{Q}}_a \mathbf{V}_{\alpha;a\leftarrow b}\tilde{\mathbf{Q}}_b \left|\psi_\alpha^+\right\rangle\right],$$
(42)

where $\tilde{\mathbf{P}}_a$ and $\tilde{\mathbf{Q}}_a$ are the projectors on $\tilde{\mathrm{P}}$ and $\tilde{\mathrm{Q}}$ in copy $a$, $\left|\phi\right\rangle$ is an eigenstate of $\mathbf{h}_\alpha^{\tilde{\mathrm{P}}}$ with eigenvalue $\varepsilon_\alpha$ and $\mathbf{V}_{\alpha;a\leftarrow b} =$ "$\mathbf{V}_{\alpha;a-b}$" implements the Fourier components of $\mathbf{V}_\alpha(\varphi) = \sum_a \mathbf{V}_{\alpha;a}e^{-ia\varphi}$ in the extended Hilbert space.

We will now connect this to our discussion of the static case, Section 4, by defining the subspaces P as the subspace of the incoming particle

$$\mathbf{P} = \tilde{\mathbf{P}}_0,$$
(43)

and Q as all the remaining subspaces:

$$\mathbf{Q} = \sum_a \tilde{\mathbf{Q}}_a + \sum_{a\neq 0} \tilde{\mathbf{P}}_a.$$
(44)

Note that Q contains all the copies of $\tilde{\mathrm{Q}}$, but also all those $\tilde{\mathrm{P}}_a$, where $a \neq 0$.

First, let us compute the Greens operator Eq. (30) in the limit $\alpha \to 0$. This can be done by fixing a wavefunction $\tilde{\mathbf{P}}\left|\psi_\alpha^+\right\rangle$ and computing the corresponding wavefunction in $\mathbf{Q}$. For $a \neq 0$ and $a \neq A$ respectively, we immediately find:

$$\tilde{\mathbf{P}}_a \left|\psi_\alpha^+\right\rangle = \frac{\alpha}{a\omega_0}\left[\tilde{\mathbf{P}}_a \mathbf{V}_{0;a\leftarrow 0}\tilde{\mathbf{P}}_0 \left|\psi_\alpha^+\right\rangle + \tilde{\mathbf{P}}_a \mathbf{V}_{0;a\leftarrow A}\tilde{\mathbf{Q}}_A \left|\psi_\alpha^+\right\rangle\right] + \mathcal{O}(\alpha^2),$$
(45)

$$\tilde{\mathbf{Q}}_a \left|\psi_\alpha^+\right\rangle = \frac{\alpha}{(a-A)\omega_0}\left[\tilde{\mathbf{Q}}_a \mathbf{V}_{0;a\leftarrow 0}\tilde{\mathbf{P}}_0 \left|\psi_\alpha^+\right\rangle + \tilde{\mathbf{Q}}_a \mathbf{V}_{0;a\leftarrow A}\tilde{\mathbf{Q}}_A \left|\psi_\alpha^+\right\rangle\right] + \mathcal{O}(\alpha^2).$$
(46)

The subspace $\tilde{\mathrm{Q}}_A$ has to be treated separately, since the denominator $a-A$ vanishes:

$$\tilde{\mathbf{Q}}_A \left|\psi_\alpha^+\right\rangle = \frac{1}{-\mathbf{h}_0^{\tilde{\mathrm{Q}}} + i\eta}\left[\tilde{\mathbf{Q}}_A \mathbf{V}_{0;A\leftarrow 0}\tilde{\mathbf{P}}_0 \left|\psi_\alpha^+\right\rangle + \tilde{\mathbf{Q}}_A \mathbf{V}_{0;A\leftarrow A}\tilde{\mathbf{Q}}_A \left|\psi_\alpha^+\right\rangle\right] + \mathcal{O}(\alpha),$$
(47)

which can be solved:

$$\tilde{\mathbf{Q}}_A \left|\psi_\alpha^+\right\rangle = \frac{1}{-\mathbf{h}_0^{\tilde{\mathrm{Q}}} - \tilde{\mathbf{Q}}_A \mathbf{V}_{0;A\leftarrow A}\tilde{\mathbf{Q}}_A + i\eta}\tilde{\mathbf{Q}}_A \mathbf{V}_{0;A\leftarrow 0}\tilde{\mathbf{P}}_0 \left|\psi_\alpha^+\right\rangle + \mathcal{O}(\alpha).$$
(48)

Comparing with the definition of the Greens operator Eq. (30) we can now identify at lowest order $\mathcal{O}(\alpha^0)$

$$\mathbf{G}_0^{\mathrm{Q}} = \tilde{\mathbf{Q}}_A \frac{1}{-\mathbf{h}_0^{\tilde{\mathrm{Q}}} - \tilde{\mathbf{Q}}_A \mathbf{V}_{0;A\leftarrow A}\tilde{\mathbf{Q}}_A + i\eta}\tilde{\mathbf{Q}}_A.$$
(49)

Using this result we can compute the effective potential $\mathbf{V}_\alpha^{\mathrm{eff}} = \mathbf{V}_0^{\mathrm{eff}} + \alpha\mathbf{V}_0^{\mathrm{eff}'} + \mathcal{O}(\alpha^2)$ from Eq. (33), where

$$\mathbf{V}_0^{\mathrm{eff}} = \tilde{\mathbf{P}}_0 \mathbf{V}_{0;0\leftarrow A}\tilde{\mathbf{Q}}_A \mathbf{G}_0^{\mathrm{Q}}\tilde{\mathbf{Q}}_A \mathbf{V}_{0;A\leftarrow 0}\tilde{\mathbf{P}}_0,$$
(50)

and $\mathbf{V}_0^{\text{eff}'}$ is a complicated expression, which we will evaluate later (see Appendix F). Note that Eq. (50) only takes into account the two resonantly coupled bands $\tilde{P}_0$ and $\tilde{Q}_A$. The projector $\mathbf{K}$ is defined as the projector on the space on which $\mathbf{V}_0^{\text{eff}}$ acts. Similarly to Lemma 1 one can establish that $\mathbf{K} = \mathbf{K}^{\text{P}}$, where in this case $\mathbf{K}^{\text{P}}$ is the projector on the space where $\mathbf{V}_{0;A\leftarrow 0}$ acts, i.e. $\mathbf{V}_{0;A\leftarrow 0}\mathbf{K}^{\text{P}} = \mathbf{V}_{0;A\leftarrow 0}$ and $\mathbf{V}_{0;A\leftarrow 0}(\tilde{\mathbf{P}}_0 - \mathbf{K}^{\text{P}}) = 0$.

The following theorem establishes a version of Theorem 2 for driven impurities and shows that the excitations from $\tilde{\text{P}}$ to $\tilde{\text{Q}}$ are suppressed in the limit $\alpha \to 0$:

**Theorem 3** *Consider a scattering problem with uncoupled Hamiltonian*

$$\mathbf{H}_0 = \tilde{\mathbf{P}}\mathbf{h}_\alpha^{\tilde{\text{P}}}\tilde{\mathbf{P}} + A\omega(\alpha)\tilde{\mathbf{Q}} + \frac{1}{\alpha}\tilde{\mathbf{Q}}\mathbf{h}_\alpha^{\tilde{\text{Q}}}\tilde{\mathbf{Q}}, \tag{51}$$

*where $\tilde{\mathbf{P}}$ and $\tilde{\mathbf{Q}}$ are complementary projectors $\tilde{\mathbf{P}} + \tilde{\mathbf{Q}} = 1$, $A \in \mathbb{Z}$ and $\frac{1}{\omega(\alpha)} = \frac{\alpha^2}{\omega_0} + \mathcal{O}(\alpha^3)$. The impurity is described by a time-periodic hermitian potential $\frac{1}{\alpha}\mathbf{V}_\alpha(\omega(\alpha)t) = \frac{1}{\alpha}\sum_a \mathbf{V}_{\alpha,a}e^{-ia\omega(\alpha)t}$, with resonant driving frequency $\omega(\alpha)$. The incoming state $|\phi\rangle$ is an eigenstate of $\mathbf{h}_\alpha^{\text{P}}$ with eigenvalue $\varepsilon_\alpha$. Assume that $\mathbf{h}_\alpha^{\text{P}}$ has a finite limit for $\alpha \to 0$ and that $\mathbf{G}_\alpha^{\text{Q}} = \tilde{\mathbf{Q}}_A\mathbf{G}_0^{\text{Q}}\tilde{\mathbf{Q}}_A + \alpha\mathbf{G}_0^{\text{Q}'} + \mathcal{O}(\alpha^2)$ (see Eq. (49)) and $\mathbf{V}_{\alpha;a} = \mathbf{V}_{0;a} + \alpha\mathbf{V}_{0;a}' + \mathcal{O}(\alpha^2)$ have a Taylor expansion at least up to first order.*

*In case that $\frac{1}{-\mathbf{h}_0^{\tilde{\text{Q}}} - \tilde{\mathbf{Q}}\mathbf{V}_{0;0}\tilde{\mathbf{Q}} + i\eta} \neq 0$, $\tilde{\mathbf{P}}\mathbf{V}_{0;0}\tilde{\mathbf{P}} = 0$, $\tilde{\mathbf{Q}}\mathbf{V}_{0;A}\tilde{\mathbf{P}} \neq 0$ and the technical assumptions (a), (b) and (c) hold, we find in the limit $\alpha \to 0$:*

(i) $\mathbf{K}|\psi_\alpha^+\rangle \to 0$.

(iv) $\tilde{\mathbf{Q}}_A|\psi_\alpha^+\rangle \to 0$. *Furthermore $\tilde{\mathbf{Q}}_a|\psi_\alpha^+\rangle \to 0$ and $\tilde{\mathbf{P}}_a|\psi_\alpha^+\rangle \to 0$, for $a \neq 0$.*

*Furthermore, if (d) and (e) hold, one can explicitly compute the limiting wavefunction using (ii) and (iii) of Theorem 1. For this one can use the following simplified expression:*

$$\bar{\mathbf{K}}\mathbf{V}_0^{\text{eff}'}\bar{\mathbf{K}} = \bar{\mathbf{K}}\mathbf{V}_0^{\text{eff}'}\mathbf{L} = \bar{\mathbf{K}}\mathbf{V}_{0;0}'\bar{\mathbf{K}} + \sum_{a\neq 0}\frac{\bar{\mathbf{K}}\mathbf{V}_{0;-a}\tilde{\mathbf{P}}\mathbf{V}_{0;a}\bar{\mathbf{K}}}{a\omega_0} + \sum_{a\neq A}\frac{\bar{\mathbf{K}}\mathbf{V}_{0;-a}\tilde{\mathbf{Q}}\mathbf{V}_{0;a}\bar{\mathbf{K}}}{(a-A)\omega_0}, \tag{52}$$

*where $\bar{\mathbf{K}} = \tilde{\mathbf{P}} - \mathbf{K}$.*

Proof: The proof is along the lines of the proof of Theorem 2. Theorem 1 already shows (i). Then we have:

$$\tilde{\mathbf{Q}}_A|\psi_\alpha^+\rangle = \sum_b \tilde{\mathbf{Q}}_A\mathbf{G}_\alpha^{\text{Q}}\tilde{\mathbf{Q}}_b\mathbf{V}_{\alpha;b\leftarrow 0}\tilde{\mathbf{P}}_0|\psi_\alpha^+\rangle \to \tilde{\mathbf{Q}}_A\mathbf{G}_0^{\text{Q}}\tilde{\mathbf{Q}}_A\mathbf{V}_{0;A\leftarrow 0}\tilde{\mathbf{P}}_0|\psi_0^+\rangle = \tilde{\mathbf{Q}}_A\mathbf{G}_0^{\text{Q}}\tilde{\mathbf{Q}}_A\mathbf{V}_{0;A\leftarrow 0}\mathbf{K}|\psi_0^+\rangle = 0. \tag{53}$$

The other two statements $\tilde{\mathbf{Q}}_a|\psi_\alpha^+\rangle \to 0$ and $\tilde{\mathbf{P}}_a|\psi_\alpha^+\rangle \to 0$ simply follow from the fact that the Greens operator $\mathbf{G}_\alpha^{\text{Q}}$ vanishes on these subspaces for $\alpha \to 0$ (see Eq. (49)). Formula Eq. (52) is derived in Appendix F.

Theorem 3 finally establishes quite generally that the excitation of particles from a slow low-energy system $\tilde{\text{P}}$ into a fast high-energy system $\tilde{\text{Q}}$ via a resonantly driven impurity is suppressed. In the appendices we discuss two explicit examples:

In Appendix G we study the scattering of bound pairs in an attractive Fermi-Hubbard chain at a driven impurity, whose frequency coincides with the binding energy of the pair. This example is along the lines of our initial discussion Section 2 on strongly interacting systems. In particular, this system can be described via the Schrieffer-Wolff transformation. In this case, Theorem 3 shows that the bound pair will not break at the impurity. Furthermore, we evaluate the pair transmission through the impurity, which turns out to be a non-trivial result.

The results of this paper also go beyond strongly interacting systems. We demonstrate this in Appendix H, where we consider a 1D optical lattice in the deep lattice limit. This limit

justifies the approximation of an optical lattice by a tight-binding chain, which is often an important starting point for Floquet-engineering quantum systems in optical lattices. This tight-binding approximation only takes the lowest energy band of the optical lattice into account. In Appendix H we add a driven impurity into the optical lattice, whose driving frequency is at resonance with the second lowest energy band. While the relative scalings of the energy bands in the deep lattice limit is quite different from the Schrieffer-Wolff transformation, by a suitable definition of $\alpha$ one can still apply Theorem 3. It shows that particles will not get excited from the lowest band into the higher band. For a simple impurity, which only acts on one lattice site, we also derive the transmission of particles through the impurity: in the limit the impurity becomes impenetrable, i.e. all particles are reflected. Note that this result is substantially different from the case of non-resonant driving: in this case the impurity becomes fully transparent, i.e. all particles are transmitted, in the same high-frequency regime.

## 6 Conclusion

In this work we studied systems where a low-energy band P is coupled by a resonantly driven impurity to a high-energy band Q. While for resonant bulk driving, the low-energy band hybridizes with the high-energy band and is effectively destroyed, we surprisingly find that for driven impurities, excitations from the low-energy band into the high-energy band are suppressed. Therefore, the low-energy description stays intact. The reason for this behaviour is that, due to a separation of time-scales between P and Q, slow low-energy particles in P cannot enter the fast impurity (which is hybridized with the high-energy band Q).

These results are based on a study of static impurities coupling two subsystems P and Q using the Lippmann-Schwinger formalism in the singular limit where the ratio of time-scales (or bandwidths) of P and Q diverges, $\alpha = \frac{\text{bandwidth of low-energy band}}{\text{bandwidth of high-energy band}} \to 0$. We established that the scattering between P and Q is suppressed and also give a formula to compute the scattering wavefunction. This is a general result in its own right and can be applied to a vast range of systems, for instance there is no restrictions on the dimension of the system, the number of particles, the shape of the impurity and the strength of the impurity (in fact the result is non-perturbative in impurity strength). The only important ingredient a clear separation of time-scales between P and Q.

The generality of the result carries over to the Floquet-driving case, which can be reduced to the static case using the extended Hilbert space formalism. We apply the result to two explicit examples. First, the breaking of bound pairs in an attractive Fermi-Hubbard chain at a driven impurity, whose frequency is at resonance with the binding energy of the pair. The second example is a driven impurity in a deep 1D optical lattice, whose driving frequency is at resonance between the lowest energy band and second lowest energy band. Especially the latter example is physically important, since optical lattices are a common way to generate tight-binding models in the lab (see for instance [1, 9, 52]). The lowest energy band of a deep optical lattice is described by tight-binding Hamiltonian, which is the starting point for many implementations of lattice systems with interesting physical properties using additional Floquet-driving (often in the high-frequency regime). The results of this paper show that in case the high frequency is at resonance with the next energy band, excitations to the higher band at a driven impurity are suppressed. In addition, the impurity becomes fully reflective due to the presence of the other band. The case of resonant driving is therefore clearly distinct from the case of non-resonant driving, where the impurity becomes transparent.

It would be interesting to check these predictions in an experiment. In particular, we think it should be possible to realize our second example, an impurity that is driven at resonance with the band gap of the two lowest bands in a deep 1D optical lattice, in an actual optical

lattice. For instance, one could start with an initial state where all the particles are confined to the left side of the impurity. After releasing the system, one can observe the number of particles that are transmitted through the impurity. We predict the following behaviour as a function of driving frequency $\omega$: if the driving frequency is not at resonance with the high-energy band, then the impurity has a high transmission. On the other hand, if the driving frequency is at resonance with the high-energy band then the transmission should decrease significantly. Also, in both cases the particles should not absorb energy from the driving and remain in the lowest energy band (this is automatically true for non-resonant driving, but also – as we show in this paper – approximately true for resonant driving). If it is possible to study this effect experimentally it would also be interesting to further investigate it theoretically, for instance taking interactions between particles into account or studying the crossover between the non-resonant and resonant driving in more detail.

The results of this paper so far are restricted to the case where the systems show a clear separation of scale in the limit $\alpha \to 0$. However, their derivation is solely based on Taylor expansions. This indicates that it should be possible to extend the results into a full perturbative series in $\alpha$. For instance, evaluating the first order correction in $\alpha$ would allow to predict the amount of particles that get excited from the low-energy band into the high-energy band. We believe that investigating this expansion provides an interesting line of research and would provide a theoretical approach to resonantly driven impurities for which non-perturbative methods (like the non-equilibrium Green's function method [39–41]) are not available. By evaluating the first few terms of this expansion this would give predictions for the behaviour of resonantly driven impurities even in cases where $\alpha$ is finite, say $\alpha \approx 0.3$. It would be interesting to study whether this perturbative series is convergent or only asymptotic (like the high-frequency expansion), which is important to understand the quality of the approximation. In either case we expect that one should still be able to observe some amount of suppression of the scattering between both bands, even for small but finite $\alpha$. Understanding this suppression could be relevant for experimental applications, for instance, if one actively tries to excite particles from one band into another band in a localized region using Floquet-driving. From a theoretical physics viewpoint the expansion would also be interesting since it is non-perturbative in impurity strength $\lambda$: as we observed in this paper, the limit $\alpha \to 0$ does not commute with the weak impurity strength $\lambda \to 0$ limit, meaning that this perturbative expansion is able to capture the regime $\alpha \ll \lambda$, which is out of reach of the standard Born series for weak couplings (which applies for $\lambda \ll \alpha$) [35].

# Acknowledgments

I would like to thank Corinna Kollath and Ameneh Sheikhan for their supervision and their helpful discussions. I would like to thank Benjamin Doyon and Kilian Bönisch for reading earlier drafts of this paper and giving useful comments. I would like to thank Paul Schindler for reading and giving feedback on the introduction.

**Funding information**   I acknowledge funding from the Deutsche Forschungsgemeinschaft (DFG, German Research Foundation) in particular under project number 277625399 - TRR 185 (B3,B4).

# A  Explicit computation in the toy model

For the toy model it is possible to solve the time-independent Schrödinger equation of the Hamiltonian Eq. (14) directly, without any need to use the Lippmann-Schwinger formalism. We make the ansatz for the scattering state with an incoming particle with momentum $k > 0$ and energy $\varepsilon$ in P:

$$\langle n_{\mathrm{P}}|\psi^+\rangle = \begin{cases} e^{ikn} + r^{\mathrm{P}} e^{-ikn}, & n \leq 0, \\ t^{\mathrm{P}} e^{ikn}, & n \geq 0, \end{cases} \tag{A.1}$$

$$\langle n_{\mathrm{Q}}|\psi^+\rangle = \begin{cases} r^{\mathrm{Q}} e^{-ik^{\mathrm{Q}} n}, & n \leq 0, \\ t^{\mathrm{Q}} e^{ik^{\mathrm{Q}} n}, & n \geq 0, \end{cases} \tag{A.2}$$

where $-\frac{2J}{\alpha}\cos k^{\mathrm{Q}} = \varepsilon = -2J\cos k$. Note that for small $\alpha$ we have $k^{\mathrm{Q}} = \frac{\pi}{2} - \alpha\cos k + \mathcal{O}(\alpha^2)$. For consistency of the ansatz at $n = 0$ we find $1 + r^{\mathrm{P}} = t^{\mathrm{P}}$ and $r^{\mathrm{Q}} = t^{\mathrm{Q}}$. The Schrödinger equation at $n = 0$ in P and Q reads:

$$-J\langle -1_{\mathrm{P}}|\psi^+\rangle - J\langle 1_{\mathrm{P}}|\psi^+\rangle + \tfrac{\lambda}{\alpha}\langle 0_{\mathrm{Q}}|\psi^+\rangle = \varepsilon\langle 0_{\mathrm{P}}|\psi^+\rangle, \tag{A.3}$$

$$-\tfrac{J}{\alpha}\langle -1_{\mathrm{Q}}|\psi^+\rangle - \tfrac{J}{\alpha}\langle 1_{\mathrm{Q}}|\psi^+\rangle + \tfrac{\lambda}{\alpha}\langle 0_{\mathrm{P}}|\psi^+\rangle = \varepsilon\langle 0_{\mathrm{Q}}|\psi^+\rangle. \tag{A.4}$$

Inserting the above ansatz and simplifying both equations gives:

$$t^{\mathrm{Q}} = \tfrac{\alpha}{\lambda} 2iJ\sin k (t^{\mathrm{P}} - 1), \tag{A.5}$$

$$t^{\mathrm{Q}} = \frac{\lambda}{2iJ\sin k^{\mathrm{Q}}} t^{\mathrm{P}}. \tag{A.6}$$

Setting both expressions equal we can solve for $t^{\mathrm{P}}$:

$$t^{\mathrm{P}} = \frac{2i\frac{\alpha}{\lambda}J\sin k}{2i\frac{\alpha}{\lambda}J\sin k - \frac{\lambda}{2iJ\sin k^{\mathrm{Q}}}} = \frac{1}{1 + \frac{\lambda^2}{4\alpha J^2 \sin k \sin k^{\mathrm{Q}}}}. \tag{A.7}$$

This result coincides with Eq. (21) since $t^{\mathrm{P}} = \langle 0_{\mathrm{P}}|\psi^+\rangle$. The probability that the particle is transmitted into the right side of P is simply given:

$$T^{\mathrm{P}} = |t^{\mathrm{P}}|^2 = \frac{1}{\left(1 + \frac{\lambda^2}{4\alpha J^2 \sin k \sin k^{\mathrm{Q}}}\right)^2}. \tag{A.8}$$

To compute the probability that the particle ends up in chain Q we need to multiply $|t^{\mathrm{Q}}|^2$ by the ratios of incoming and outgoing velocities:

$$T^{\mathrm{P}\to\mathrm{Q}} = 2\frac{J_{\mathrm{out}}^{\mathrm{Q}}}{J_{\mathrm{in}}^{\mathrm{P}}} = 2\frac{2\frac{1}{\alpha}J\sin k^{\mathrm{Q}} |t^{\mathrm{Q}}|^2}{2J\sin k} = 2\frac{\lambda^2}{4\alpha J^2 \sin k \sin k^{\mathrm{Q}}} T^{\mathrm{P}}. \tag{A.9}$$

The extra prefactor 2 was added since the particle can escape in two directions in Q. We give a plot of $T^{\mathrm{P}}$ and $T^{\mathrm{P}\to\mathrm{Q}}$ in Fig. 3 for different values of $\alpha$. One can see that both $T^{\mathrm{P}}$ and $T^{\mathrm{P}\to\mathrm{Q}}$ vanish as $\alpha \to 0$ throughout the whole Brillouin zone. This is of course the same result as we had already found in Section 3.

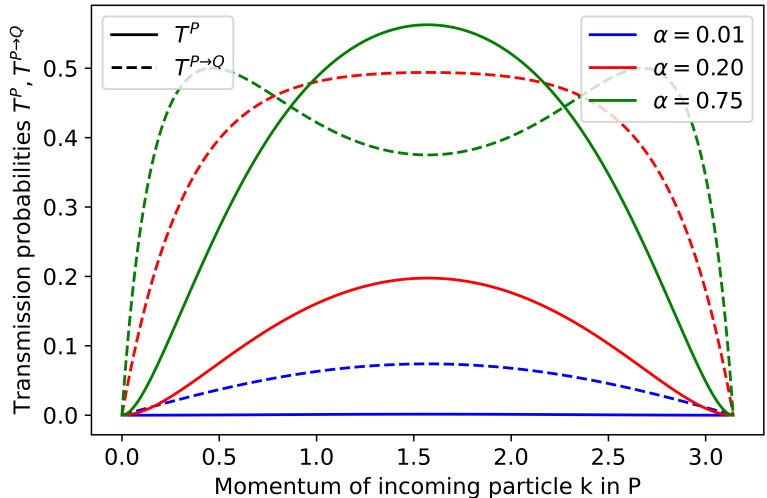

Figure 3: Transmission probabilities $T^{\mathrm{P}}$ and $T^{\mathrm{P}\to\mathrm{Q}}$ as function of particle momentum $k$ for different values of $\alpha$ ($J = \lambda = 1$).

## B Relation to Fano-resonance in Q in the toy model

In the toy model (see Section 3) as $\alpha \to 0$ we do not only observe the suppression of scattering from P to Q, but also a Fano-resonance for scattering in Q, i.e. the transmission probability for an incoming particle in Q vanishes at the energy of P. In this appendix we will discuss to what extent these two effects are related.

For this, let us consider an incoming particle in Q and solve the scattering problem exactly. This can be done similarly to the computations in the last section. For an incoming particle in Q the ansatz for the wavefunction is:

$$\langle n_{\mathrm{Q}}|\psi^+\rangle = \begin{cases} e^{ikn} + r^{\mathrm{Q}}e^{-ikn}, & n \leq 0, \\ t^{\mathrm{Q}}e^{ikn}, & n \geq 0, \end{cases} \tag{B.1}$$

$$\langle n_{\mathrm{P}}|\psi^+\rangle = \begin{cases} r^{\mathrm{P}}e^{-ik^{\mathrm{P}}n}, & n \leq 0, \\ t^{\mathrm{P}}e^{ik^{\mathrm{P}}n}, & n \geq 0, \end{cases} \tag{B.2}$$

where now $-2J\cos k^{\mathrm{P}} = \varepsilon = -2\frac{J}{\alpha}\cos k$. Note that $k^{\mathrm{P}}$ is only real if $|\cos k| < \alpha$, otherwise it has a positive imaginary part, indicating that transmission into P is not possible since the energies do not match. Again we have $1 + r^{\mathrm{Q}} = t^{\mathrm{Q}}$ and $r^{\mathrm{P}} = t^{\mathrm{P}}$. The Schrödinger equation can be solved similarly to above. The solution is:

$$t^{\mathrm{Q}} = \frac{1}{1 + \frac{\lambda^2}{4\alpha J^2 \sin k^{\mathrm{P}} \sin k}}, \tag{B.3}$$

$$t^{\mathrm{P}} = \frac{\lambda}{2i\alpha J \sin k^{\mathrm{P}}}t^{\mathrm{Q}}. \tag{B.4}$$

From this we find the transmission probability in Q:

$$T^{\mathrm{Q}} = \left|t^{\mathrm{Q}}\right|^2 = \frac{1}{\left|1 + \frac{\lambda^2}{4\alpha J^2 \sin k^{\mathrm{P}} \sin k}\right|^2}, \tag{B.5}$$

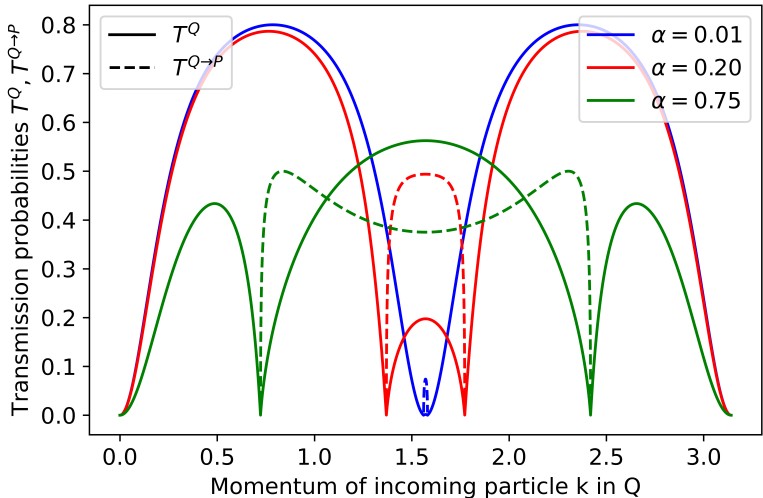

Figure 4: Transmission probabilities $T^{\mathrm{Q}}$ and $T^{\mathrm{Q}\rightarrow\mathrm{P}}$ as function of particle momentum $k$ for different values of $\alpha$ ($J = \lambda = 1$). The zeros of $T^{\mathrm{Q}}$ are located at the boundary $\cos k = \pm\alpha$ of the thinner band P. Transmission $T^{\mathrm{Q}\rightarrow\mathrm{P}}$ from Q to P only happens if the energy of the incoming particle is inside the bandwidth of P.

and the probability for the incoming particle in Q to end up in P (which only happens if the energy of the incoming particle lies inside the bandwidth of P):

$$T^{\mathrm{Q}\rightarrow\mathrm{P}} = 2\frac{J^{\mathrm{P}}_{\mathrm{out}}}{J^{\mathrm{Q}}_{\mathrm{in}}} = 2\frac{2J\sin k^{\mathrm{P}}\left|t^{\mathrm{P}}\right|^2}{2\frac{1}{\alpha}J\sin k} = 2\frac{\lambda^2}{4\alpha J^2 \sin k \sin k^{\mathrm{P}}}T^{\mathrm{Q}}. \tag{B.6}$$

We give a plot of these probabilities for different values of $\alpha$ in Fig. 4. We can see that $T^{\mathrm{Q}}$ vanishes at $\cos k = \pm\alpha$, i.e at the boundary of the band P, even for finite $\alpha$. As $\alpha \rightarrow 0$, these two zeros merge and create the Fano-resonance at $E = 0$. Inside the bandwidth of P the transmission probabilities also show a non-trivial behaviour. However, this behaviour can only be observed if the energy of the incoming particle is inside the bandwidth $|\varepsilon| < \alpha$. For very small $\alpha$ this requires precise fine tuning of the momentum of the incoming particle. Otherwise the transmission profile will correspond to the Fano-resonance observed for energies outside of the bandwidth of P. We conclude that, as $\alpha \rightarrow 0$ it is not possible to infer detailed information about system P from measuring the scattering of incoming particles in Q. This is simply due to the separation of time/energy-scales. From the perspective of Q the continuum of energies in P collapses to a point $E = 0$ and is thus not resolvable for a particle in Q.

However, for the toy model there is still a way to illustrate the connection between the Fano-resonance in Q and the vanishing of the scattering from P to Q. The Fano-resonance happens because $t^{\mathrm{Q}}$ in Eq. (B.3) vanishes at $\varepsilon = 0$ as $\alpha \rightarrow 0$. This means that the wavefunction vanishes at site 0 in Q. Because the wavefunction vanishes at site 0 it is impossible for a particle to reach the other side, i.e. it has to be reflected. This is similar to the artificial boundary condition we found in system P and for the toy model means that the local density of states vanishes at site 0 in Q. Now, because an incoming particle in P would need to pass through site 0 in Q, a vanishing local density of states at that site prevents scattering from the slow system P to the fast system Q. Therefore, the Fano-resonance in Q can explain why the scattering from P to Q is suppressed as $\alpha \rightarrow 0$.

We would like to conclude this discussion by emphasising that, while it appears in the toy model, in general a Fano-resonance does not need to appear in Q as $\alpha \rightarrow 0$. For instance, if in

the toy model the coupling between both chains is replaced by:

$$\lambda\left(|0_P\rangle\langle 0_Q| + |0_Q\rangle\langle 0_P|\right) \to \lambda\left(|0_P\rangle\frac{\langle 0_Q| - \langle 1_Q|}{\sqrt{2}} + \frac{|0_Q\rangle - |1_Q\rangle}{\sqrt{2}}\langle 0_P|\right), \tag{B.7}$$

there is no Fano-resonance in Q at energy $E = 0$ as $\alpha \to 0$. Therefore, the result of this paper is more general than the Fano-resonance in Q and in particular a Fano-resonance cannot be used to understand the physical effect behind the suppression of the scattering from P to Q.

## C   Proof of Theorem 1

The Lippmann-Schwinger equation Eq. (32) is an implicit equation for the infinite dimensional vector $\mathbf{P}|\psi_\alpha^+\rangle$, which we cannot solve directly. However, we can expect that in the limit $\alpha \to 0$ the right hand side only depends on the much smaller (and often even finite dimensional) vector $\mathbf{K}|\psi_\alpha^+\rangle$, since all other parts are annihilated by the potential $\mathbf{V}_0^{\text{eff}}$. For example in the toy model Eq. (19), the right hand side only depends on $\langle 0_P|\mathbf{P}|\psi^+\rangle$ and thus $\mathbf{K} = |0_P\rangle\langle 0_P|$.

As a first step split Eq. (32) into two coupled equations for $\mathbf{K}$ and $\bar{\mathbf{K}}$:

$$\mathbf{K}|\psi_\alpha^+\rangle = \mathbf{K}|\phi\rangle + \frac{1}{\alpha}\mathbf{K}\mathbf{G}_\alpha^P\mathbf{V}_\alpha^{\text{eff}}\mathbf{K}|\psi_\alpha^+\rangle + \frac{1}{\alpha}\mathbf{K}_\alpha^P\mathbf{V}_\alpha^{\text{eff}}\bar{\mathbf{K}}|\psi_\alpha^+\rangle, \tag{C.1}$$

$$\bar{\mathbf{K}}|\psi_\alpha^+\rangle = \bar{\mathbf{K}}|\phi\rangle + \frac{1}{\alpha}\bar{\mathbf{K}}\mathbf{G}_\alpha^P\mathbf{V}_\alpha^{\text{eff}}\mathbf{K}|\psi_\alpha^+\rangle + \frac{1}{\alpha}\bar{\mathbf{K}}\mathbf{G}_\alpha^P\mathbf{V}_\alpha^{\text{eff}}\bar{\mathbf{K}}|\psi_\alpha^+\rangle. \tag{C.2}$$

Using technical assumption (a) we can solve Eq. (C.2):

$$\bar{\mathbf{K}}|\psi_\alpha^+\rangle = \frac{1}{\bar{\mathbf{K}} - \frac{1}{\alpha}\bar{\mathbf{K}}\mathbf{G}_\alpha^P\mathbf{V}_\alpha^{\text{eff}}\bar{\mathbf{K}}}\left(\bar{\mathbf{K}}|\phi\rangle + \frac{1}{\alpha}\bar{\mathbf{K}}\mathbf{G}_\alpha^P\mathbf{V}_\alpha^{\text{eff}}\mathbf{K}|\psi_\alpha^+\rangle\right). \tag{C.3}$$

which we can insert into Eq. (C.1) and use (b) to solve for $\mathbf{K}|\psi^+\rangle$:

$$\mathbf{K}|\psi_\alpha^+\rangle = \frac{1}{\mathbf{K} - \frac{1}{\alpha}\mathbf{K}\mathbf{G}_\alpha^P\mathbf{V}_\alpha^{\text{eff}}\mathbf{K} - \frac{1}{\alpha}\mathbf{K}\mathbf{G}_\alpha^P\mathbf{V}_\alpha^{\text{eff}}\bar{\mathbf{K}}\frac{1}{\bar{\mathbf{K}} - \frac{1}{\alpha}\bar{\mathbf{K}}\mathbf{G}_\alpha^P\mathbf{V}_\alpha^{\text{eff}}\bar{\mathbf{K}}}\frac{1}{\alpha}\bar{\mathbf{K}}\mathbf{G}_\alpha^P\mathbf{V}_\alpha^{\text{eff}}\mathbf{K}}$$

$$\times\left(\mathbf{K} + \frac{1}{\alpha}\mathbf{K}\mathbf{G}_\alpha^P\mathbf{V}_\alpha^{\text{eff}}\bar{\mathbf{K}}\frac{1}{\bar{\mathbf{K}} - \frac{1}{\alpha}\bar{\mathbf{K}}\mathbf{G}_\alpha^P\mathbf{V}_\alpha^{\text{eff}}\bar{\mathbf{K}}}\bar{\mathbf{K}}\right)|\phi\rangle. \tag{C.4}$$

This result corresponds to Eq. (21) for the toy model. Now we are ready to study the limit $\alpha \to 0$. Due to the definitions of the projectors we have

$$\mathbf{G}_\alpha^P \to \mathbf{G}_0^P, \tag{C.5}$$

$$\mathbf{V}_\alpha^{\text{eff}}\mathbf{K} \to \mathbf{V}_0^{\text{eff}}\mathbf{K}, \tag{C.6}$$

$$\frac{1}{\alpha}\mathbf{V}_\alpha^{\text{eff}}\bar{\mathbf{K}} \to \mathbf{V}_0^{\text{eff}'}\bar{\mathbf{K}}, \tag{C.7}$$

which we can directly insert into Eq. (C.4):

$$\mathbf{K}|\psi_\alpha^+\rangle = -\frac{\alpha}{\mathbf{K}\mathbf{G}_0^P\mathbf{V}_0^{\text{eff}}\mathbf{K} + \mathbf{K}\mathbf{G}_0^P\mathbf{V}_0^{\text{eff}'}\bar{\mathbf{K}}\frac{1}{\bar{\mathbf{K}} - \bar{\mathbf{K}}\mathbf{G}_0^P\mathbf{V}_0^{\text{eff}'}\bar{\mathbf{K}}}\bar{\mathbf{K}}\mathbf{G}_0^P\mathbf{V}_0^{\text{eff}}\mathbf{K}} \tag{C.8}$$

$$\times\left(\mathbf{K} + \mathbf{K}\mathbf{G}_0^P\mathbf{V}_0^{\text{eff}'}\bar{\mathbf{K}}\frac{1}{\bar{\mathbf{K}} - \bar{\mathbf{K}}\mathbf{G}_0^P\mathbf{V}_0^{\text{eff}'}\bar{\mathbf{K}}}\bar{\mathbf{K}}\right)|\phi\rangle + \mathcal{O}(\alpha^2), \tag{C.9}$$

where (a) and (b) ensure that the operators are still invertible in the limit. Therefore, $\mathbf{K}\left|\psi_\alpha^+\right\rangle \to 0$ as $\alpha \to 0$ which is the first conclusion.

From Eq. (C.9) we could obtain the full wavefunction by inserting it into equation Eq. (C.3) where the $\alpha$ and $\frac{1}{\alpha}$ of both formulas cancel. The resulting equation would be correct, but quite complicated and not suitable for explicit calculations because we need to invert the infinite dimensional operator $\bar{\mathbf{K}} - \bar{\mathbf{K}}\mathbf{G}_0^{\mathrm{P}}\mathbf{V}_0^{\mathrm{eff}}\bar{\mathbf{K}}$. Interestingly there is a much simpler formula in case $\bar{\mathbf{K}}\mathbf{V}_0^{\mathrm{eff}} = 0$.

To see this let us go back to the original Lippmann-Schwinger equation Eq. (28):

$$\mathbf{P}\left|\psi_\alpha^+\right\rangle = |\phi\rangle + \frac{1}{\alpha}\mathbf{G}_\alpha^{\mathrm{P}}\mathbf{V}_\alpha^{\mathrm{eff}}\mathbf{P}\left|\psi^+\right\rangle. \tag{C.10}$$

Due to the findings in the previous step we know that in the limit $\mathbf{K}\left|\psi_\alpha^+\right\rangle = \alpha\mathbf{K}\left|\psi_0^{+\prime}\right\rangle + \mathcal{O}(\alpha^2)$, where $\mathbf{K}\left|\psi_0^{+\prime}\right\rangle$ is the complicated expression in Eq. (C.9). Expanding everything in a Taylor series and using $\mathbf{K}\left|\psi_0^+\right\rangle = 0$ we find that:

$$\mathbf{P}\left|\psi_0^+\right\rangle = |\phi\rangle + \mathbf{G}_0^{\mathrm{P}}\left(\mathbf{V}_0^{\mathrm{eff}}\mathbf{K}\left|\psi_0^{+\prime}\right\rangle + \mathbf{V}_0^{\mathrm{eff}\prime}\bar{\mathbf{K}}\left|\psi_0^+\right\rangle\right). \tag{C.11}$$

Multiply both sides by $\mathbf{K}$, use the extra assumption $\bar{\mathbf{K}}\mathbf{V}_0^{\mathrm{eff}} = 0$ and the definition of $\mathbf{L}$ (which implies $\bar{\mathbf{K}}\mathbf{V}_0^{\mathrm{eff}\prime}\bar{\mathbf{K}} = \bar{\mathbf{K}}\mathbf{V}_0^{\mathrm{eff}\prime}\mathbf{L}$) to obtain:

$$0 = \mathbf{K}\left|\psi_0^+\right\rangle = \mathbf{K}|\phi\rangle + \mathbf{K}\mathbf{G}_0^{\mathrm{P}}\mathbf{K}\left(\mathbf{K}\mathbf{V}_0^{\mathrm{eff}}\mathbf{K}\left|\psi_0^{+\prime}\right\rangle + \mathbf{K}\mathbf{V}_0^{\mathrm{eff}}\bar{\mathbf{K}}\left|\psi_0^+\right\rangle\right) + \mathbf{K}\mathbf{G}_0^{\mathrm{P}}\bar{\mathbf{K}}\mathbf{V}_0^{\mathrm{eff}\prime}\mathbf{L}\left|\psi_0^+\right\rangle. \tag{C.12}$$

From this we can first find an expression for $\mathbf{K}\mathbf{V}_0^{\mathrm{eff}}\mathbf{K}\left|\psi_0^{+\prime}\right\rangle$ using (d) and then by reinserting this into Eq. (C.11) we have:

$$\mathbf{P}\left|\psi_0^+\right\rangle = \left(1 - \mathbf{G}_0^{\mathrm{P}}\mathbf{K}\frac{1}{\mathbf{K}\mathbf{G}_0^{\mathrm{P}}\mathbf{K}}\mathbf{K}\right)\left(|\phi\rangle + \mathbf{G}_0^{\mathrm{P}}\bar{\mathbf{K}}\mathbf{V}_0^{\mathrm{eff}\prime}\mathbf{L}\left|\psi_0^+\right\rangle\right), \tag{C.13}$$

which is the third conclusion of the theorem. The second conclusion can finally be obtained from the third conclusion by applying $\mathbf{L}$ to both sides and solving for $\mathbf{L}\left|\psi^+\right\rangle$ using (e).

## D  Proof of Lemma 1

The Taylor expansion of the effective potential Eq. (33) is given by:

$$\mathbf{V}_\alpha^{\mathrm{eff}} = \mathbf{P}\mathbf{V}_0\mathbf{Q}\mathbf{G}_0^{\mathrm{Q}}\mathbf{Q}\mathbf{V}_0\mathbf{P} + \alpha\left(\mathbf{P}\mathbf{V}_0'\mathbf{P} + \mathbf{P}\mathbf{V}_0'\mathbf{Q}\mathbf{G}_0^{\mathrm{Q}}\mathbf{Q}\mathbf{V}_0\mathbf{P} + \mathbf{P}\mathbf{V}_0\mathbf{Q}\mathbf{G}_0^{\mathrm{Q}\prime}\mathbf{Q}\mathbf{V}_0\mathbf{P} + \mathbf{P}\mathbf{V}_0\mathbf{Q}\mathbf{G}_0^{\mathrm{Q}}\mathbf{Q}\mathbf{V}_0'\mathbf{P}\right) + \mathcal{O}(\alpha^2), \tag{D.1}$$

and thus

$$\mathbf{V}_0^{\mathrm{eff}} = \mathbf{P}\mathbf{V}_0\mathbf{Q}\mathbf{G}_0^{\mathrm{Q}}\mathbf{Q}\mathbf{V}_0\mathbf{P}, \tag{D.2}$$

$$\mathbf{V}_0^{\mathrm{eff}\prime} = \mathbf{P}\mathbf{V}_1\mathbf{P} + \mathbf{P}\mathbf{V}_1\mathbf{Q}\mathbf{G}_0^{\mathrm{Q}}\mathbf{Q}\mathbf{V}_0\mathbf{P} + \mathbf{P}\mathbf{V}_0\mathbf{Q}\mathbf{G}_1^{\mathrm{Q}}\mathbf{Q}\mathbf{V}_0\mathbf{P} + \mathbf{P}\mathbf{V}_0\mathbf{Q}\mathbf{G}_0^{\mathrm{Q}}\mathbf{Q}\mathbf{V}_1\mathbf{P}. \tag{D.3}$$

We need to show that $\mathbf{V}_0^{\mathrm{eff}} \neq 0$. The definitions of $\mathbf{K}^{\mathrm{P}}$ and $\mathbf{K}^{\mathrm{Q}}$ are made such that we can think of $\mathbf{K}^{\mathrm{Q}}\mathbf{V}_0\mathbf{K}^{\mathrm{P}}$ as an invertible operator. In fact, we can think about

$$\mathbf{V}_0^{\mathrm{eff}} = \mathbf{K}^{\mathrm{P}}\mathbf{V}_0\mathbf{K}^{\mathrm{Q}}\mathbf{G}_0^{\mathrm{Q}}\mathbf{K}^{\mathrm{Q}}\mathbf{V}_0\mathbf{K}^{\mathrm{P}}, \tag{D.4}$$

as a product of three invertible operators ($\mathbf{K}^{\mathrm{Q}}\mathbf{G}_0^{\mathrm{Q}}\mathbf{K}^{\mathrm{Q}}$ is invertible due to technical assumption (c)). Therefore the product is also invertible and in particular $\mathbf{V}_0^{\mathrm{eff}}$ cannot be 0. Furthermore

$\mathbf{V}_0^{\text{eff}}$ maps all vectors in the subspace described by $\mathbf{K}^{\text{P}}$ onto non-zero vectors (except the zero-vector of course). Since this is exactly the definition of $\mathbf{K}$ we find

$$\mathbf{K} = \mathbf{K}^{\text{P}}. \tag{D.5}$$

The next result is found by applying $\bar{\mathbf{K}} = \mathbf{P} - \mathbf{K}$ onto Eq. (D.4):

$$\mathbf{V}_0^{\text{eff}}\bar{\mathbf{K}} = \mathbf{V}_0^{\text{eff}}\mathbf{K}\bar{\mathbf{K}} = 0\,, \tag{D.6}$$

$$\bar{\mathbf{K}}\mathbf{V}_0^{\text{eff}} = \bar{\mathbf{K}}\mathbf{K}^{\text{P}}\mathbf{V}_0\mathbf{K}^{\text{Q}}\mathbf{G}_0^{\text{Q}}\mathbf{K}^{\text{Q}}\mathbf{V}_0\mathbf{K}^{\text{P}} = \bar{\mathbf{K}}\mathbf{K}\mathbf{V}_0\mathbf{K}^{\text{Q}}\mathbf{G}_0^{\text{Q}}\mathbf{K}^{\text{Q}}\mathbf{V}_0\mathbf{K}^{\text{P}} = 0\,. \tag{D.7}$$

While the first statement is trivially true for any operator, the second one only holds because of the sandwich structure of $\mathbf{V}_0^{\text{eff}}$ (it would be automatically true if $\mathbf{V}_0^{\text{eff}}$ was hermitian).

For the last result of the lemma apply $\bar{\mathbf{K}}$ on both sides of Eq. (D.3) and use $\mathbf{V}_0\bar{\mathbf{K}} = \bar{\mathbf{K}}\mathbf{V}_0 = 0$:

$$\bar{\mathbf{K}}\mathbf{V}_0^{\text{eff}'}\bar{\mathbf{K}} = \bar{\mathbf{K}}\Big(\mathbf{P}\mathbf{V}_0'\mathbf{P} + \mathbf{P}\mathbf{V}_0'\mathbf{Q}\mathbf{G}_0^{\text{Q}}\mathbf{Q}\mathbf{V}_0\mathbf{P} + \mathbf{P}\mathbf{V}_0\mathbf{Q}\mathbf{G}_0^{\text{Q}'}\mathbf{Q}\mathbf{V}_0\mathbf{P} + \mathbf{P}\mathbf{V}_0\mathbf{Q}\mathbf{G}_0^{\text{Q}}\mathbf{Q}\mathbf{V}_0'\mathbf{P}\Big)\bar{\mathbf{K}} \tag{D.8}$$

$$= \bar{\mathbf{K}}\mathbf{V}_0'\bar{\mathbf{K}}. \tag{D.9}$$

In fact, $\mathbf{L}$ is given as the projector onto the space where $\bar{\mathbf{K}}\mathbf{V}_0'\bar{\mathbf{K}}$ acts, i.e $\bar{\mathbf{K}}\mathbf{V}_0'\bar{\mathbf{K}} = \bar{\mathbf{K}}\mathbf{V}_0'\mathbf{L}$, which is (because $\mathbf{V}_0'$ is hermitian) also equal to $\mathbf{L}\mathbf{V}_0'\bar{\mathbf{K}}$. From this we conclude $\bar{\mathbf{K}}\mathbf{V}_0'\bar{\mathbf{K}} = \mathbf{L}\mathbf{V}_0'\mathbf{L}$ which finishes the proof of the lemma.

# E   Discussion of the technical assumptions

The technical assumptions which are necessary for the proofs are the following:

(a) $\bar{\mathbf{K}} - \frac{1}{\alpha}\bar{\mathbf{K}}\mathbf{G}_\alpha^{\text{P}}\mathbf{V}_\alpha^{\text{eff}}\bar{\mathbf{K}}$ is invertible for all $\alpha$ and its limit $\bar{\mathbf{K}} - \bar{\mathbf{K}}\mathbf{G}_0^{\text{P}}\mathbf{V}_0^{\text{eff}'}\bar{\mathbf{K}}$ is also invertible.

(b) $\mathbf{K} - \frac{1}{\alpha}\mathbf{K}\mathbf{G}_\alpha^{\text{P}}\mathbf{V}_\alpha^{\text{eff}}\mathbf{K} - \frac{1}{\alpha}\mathbf{K}\mathbf{G}_\alpha^{\text{P}}\mathbf{V}_\alpha^{\text{eff}}\bar{\mathbf{K}}\frac{1}{\bar{\mathbf{K}} - \frac{1}{\alpha}\bar{\mathbf{K}}\mathbf{G}_\alpha^{\text{P}}\mathbf{V}_\alpha^{\text{eff}}\bar{\mathbf{K}}}\frac{1}{\alpha}\bar{\mathbf{K}}\mathbf{G}_\alpha^{\text{P}}\mathbf{V}_\alpha^{\text{eff}}\mathbf{K}$ is invertible for all $\alpha$ and its limit $\mathbf{K}\mathbf{G}_0^{\text{P}}\mathbf{V}_0^{\text{eff}}\mathbf{K} + \mathbf{K}\mathbf{G}_0^{\text{P}}\mathbf{V}_0^{\text{eff}'}\bar{\mathbf{K}}\frac{1}{\bar{\mathbf{K}} - \bar{\mathbf{K}}\mathbf{G}_0^{\text{P}}\mathbf{V}_0^{\text{eff}'}\bar{\mathbf{K}}}\bar{\mathbf{K}}\mathbf{G}_0^{\text{P}}\mathbf{V}_0^{\text{eff}}\mathbf{K}$ is also invertible.

(c) $\mathbf{K}^{\text{Q}}\mathbf{G}_0^{\text{Q}}\mathbf{K}^{\text{Q}}$ is invertible.

(d) $\mathbf{K}\mathbf{G}_0^{\text{P}}\mathbf{K}$ is invertible.

(e) $\mathbf{L} - \Big(\mathbf{L} - \mathbf{L}\mathbf{G}_0^{\text{P}}\mathbf{K}\frac{1}{\mathbf{K}\mathbf{G}_0^{\text{P}}\mathbf{K}}\mathbf{K}\Big)\mathbf{K}\mathbf{G}_0^{\text{P}}\bar{\mathbf{K}}\mathbf{V}_0^{\text{eff}'}\mathbf{L}$ is invertible.

They state that certain operators are invertible. In the following we would like to motivate why we regard these assumptions as technical, i.e. why we expect them to hold in almost every system. For simplicity let us restrict to the case where the potential has only finite range, i.e. it only acts on a finite subspace of $\mathbf{P}$. Then the subspaces described by $\mathbf{K}$ and $\mathbf{L}$ are actually finite dimensional and thus we can think of the operators in (b), (c), (d) and (e) as finite dimensional matrices. The operator $\bar{\mathbf{K}} - \frac{1}{\alpha}\bar{\mathbf{K}}\mathbf{G}_\alpha^{\text{P}}\mathbf{V}_\alpha^{\text{eff}}\bar{\mathbf{K}}$ in (a) is still infinite dimensional, however outside of where the potential acts it is simply a unit operator, which can be trivially inverted. Therefore also the operator in (a) can be thought of as a finite dimensional matrix. Now our intuition tells us that a generic finite dimensional matrix is invertible. Especially note that the matrices in (a),(b) and (e) depend non-linearly on the potential $\mathbf{V}_\alpha^{\text{eff}}$. Thus, even if there would be a specific situation where they are not invertible, even a tiny change in for example the potential strength should make them invertible.

The argument for (c) and (d) is similar, but a bit different because the matrices representing the Greens operators will not be generic. Instead they are usually block-diagonal, where the

blocks correspond to different symmetry sectors (like parity) or pairwise independent sub-subsystems of either P and Q. We can however expect that each of these blocks is a generic matrix, which in turn should be invertible. Thus, also the whole matrix is invertible. There is an important exception to that, namely when one of these blocks completely vanishes in the limit $\alpha \to 0$. This can happen when a subsystem of Q has a Hamiltonian which diverges as $\alpha \to 0$, i.e. it evolves even faster than the other parts in Q. Since the block is identically zero it is not invertible and thus (c) does not hold. In that case, however, one can still use the results of Theorem 1 to solve the whole problem.

## F Derivation of equation (52) in proof of Theorem 3

Similarly to the derivation of the zeroth order Greens operator Eq. (49) one can compute the next order corrections for $a \neq 0$ and $a \neq A$ respectively:

$$\tilde{\mathbf{P}}_a \mathbf{G}_0^{Q'} \tilde{\mathbf{P}}_a = \frac{1}{a\omega_0}, \qquad \tilde{\mathbf{P}}_a \mathbf{G}_0^{Q'} \tilde{\mathbf{Q}}_A = \frac{1}{a\omega_0} \tilde{\mathbf{P}}_a \mathbf{V}_{0;a\leftarrow A} \tilde{\mathbf{Q}}_A \mathbf{G}_0^Q \tilde{\mathbf{Q}}_A, \qquad (F.1)$$

$$\tilde{\mathbf{Q}}_a \mathbf{G}_0^{Q'} \tilde{\mathbf{P}}_a = \frac{1}{(a-A)\omega_0}, \qquad \tilde{\mathbf{Q}}_a \mathbf{G}_0^{Q'} \tilde{\mathbf{Q}}_A = \frac{1}{(a-A)\omega_0} \tilde{\mathbf{Q}}_a \mathbf{V}_{0;a\leftarrow A} \tilde{\mathbf{Q}}_A \mathbf{G}_0^Q \tilde{\mathbf{Q}}_A. \qquad (F.2)$$

All other parts of $\mathbf{G}_0^{Q'}$ vanish except for $\tilde{\mathbf{Q}}_A \mathbf{G}_0^{Q'} \tilde{\mathbf{Q}}_A$, which is given by a complicated expression (which we will not need). From this one can derive the following complicated expression for the first order correction to the effective potential Eq. (33):

$$\begin{aligned}
\mathbf{V}_0^{\text{eff}'} = & \; \tilde{\mathbf{P}}_0 \mathbf{V}'_{0;0\leftarrow 0} \tilde{\mathbf{P}}_0 \\
& + \tilde{\mathbf{P}}_0 \mathbf{V}'_{0;0\leftarrow A} \tilde{\mathbf{Q}}_A \mathbf{G}_0^Q \tilde{\mathbf{Q}}_A \mathbf{V}_{0;A\leftarrow 0} \tilde{\mathbf{P}}_0 + \tilde{\mathbf{P}}_0 \mathbf{V}_{0;0\leftarrow A} \tilde{\mathbf{Q}}_A \mathbf{G}_0^{Q'} \tilde{\mathbf{Q}}_A \mathbf{V}_{0;A\leftarrow 0} \tilde{\mathbf{P}}_0 + \tilde{\mathbf{P}}_0 \mathbf{V}_{0;0\leftarrow A} \tilde{\mathbf{Q}}_A \mathbf{G}_0^Q \tilde{\mathbf{Q}}_A \mathbf{V}'_{0;A\leftarrow 0} \tilde{\mathbf{P}}_0 \\
& + \sum_{a\neq 0} \tilde{\mathbf{P}}_0 \mathbf{V}_{0;0\leftarrow a} \tilde{\mathbf{P}}_a \mathbf{G}_0^{Q'} \tilde{\mathbf{P}}_a \mathbf{V}_{0;a\leftarrow 0} \tilde{\mathbf{P}}_0 + \sum_{b\neq A} \tilde{\mathbf{P}}_0 \mathbf{V}_{0;0\leftarrow a} \tilde{\mathbf{Q}}_a \mathbf{G}_0^{Q'} \tilde{\mathbf{Q}}_a \mathbf{V}_{0;a\leftarrow 0} \tilde{\mathbf{P}}_0 \\
& + \sum_{a\neq A} \tilde{\mathbf{P}}_0 \mathbf{V}_{0;0\leftarrow a} \tilde{\mathbf{Q}}_a \mathbf{G}_0^{Q'} \tilde{\mathbf{Q}}_A \mathbf{V}_{0;A\leftarrow 0} \tilde{\mathbf{P}}_0.
\end{aligned} \qquad (F.3)$$

Fortunately, in order to apply formulas (ii) and (iii) of Theorem 1 we do not need to compute $\mathbf{V}_0^{\text{eff}'}$, but it is sufficient to compute $\bar{\mathbf{K}} \mathbf{V}_0^{\text{eff}'} \bar{\mathbf{K}}$. Recall that $\bar{\mathbf{K}} = \tilde{\mathbf{P}}_0 - \mathbf{K}$ and that $\mathbf{V}_{0;A\leftarrow 0}\mathbf{K} = \mathbf{V}_{0;A\leftarrow 0}\tilde{\mathbf{P}}_0$. This implies $\mathbf{V}_{0;A\leftarrow 0}\bar{\mathbf{K}} = 0$ and $\bar{\mathbf{K}}\mathbf{V}_{0;0\leftarrow A} = \left(\mathbf{V}_{0;A\leftarrow 0}\bar{\mathbf{K}}\right)^\dagger = 0$, which allows us to drastically simplify:

$$\bar{\mathbf{K}} \mathbf{V}_0^{\text{eff}'} \bar{\mathbf{K}} = \bar{\mathbf{K}} \mathbf{V}'_{0;0\leftarrow 0} \bar{\mathbf{K}} + \sum_{a\neq 0} \bar{\mathbf{K}} \mathbf{V}_{0;0\leftarrow b} \tilde{\mathbf{P}}_a \mathbf{G}_0^{Q'} \tilde{\mathbf{P}}_a \mathbf{V}_{0;a\leftarrow 0} \bar{\mathbf{K}} + \sum_{a\neq A} \bar{\mathbf{K}} \mathbf{V}_{0;0\leftarrow b} \tilde{\mathbf{Q}}_a \mathbf{G}_0^{Q'} \tilde{\mathbf{Q}}_a \mathbf{V}_{0;a\leftarrow 0} \bar{\mathbf{K}} \qquad (F.4)$$

$$= \bar{\mathbf{K}} \mathbf{V}'_{0;0\leftarrow 0} \bar{\mathbf{K}} + \sum_{a\neq 0} \frac{\bar{\mathbf{K}} \mathbf{V}_{0;0\leftarrow a} \tilde{\mathbf{P}}_a \mathbf{V}_{0;a\leftarrow 0} \bar{\mathbf{K}}}{a\omega_0} + \sum_{a\neq A} \frac{\bar{\mathbf{K}} \mathbf{V}_{0;0\leftarrow a} \tilde{\mathbf{Q}}_a \mathbf{V}_{0;a\leftarrow 0} \bar{\mathbf{K}}}{(a-A)\omega_0}. \qquad (F.5)$$

This is the result in the extended Hilbert space. Expressing it in terms of operators in the original system gives Eq. (52).

## G Application: Scattering of bound pairs at a resonantly Floquet-driven impurity

In this appendix we apply Theorem 3 to an explicit example: bound pairs in an attractive Fermi-Hubbard chain scattering at a driven impurity, which is at resonance with the binding

energy of the pair. This system can be studied using the Schrieffer-Wolff transformation and is an example for our original motivation to study these systems, see Section 2. An analysis of the non-resonant case for the same driven impurity in exactly the same regime has been carried out recently [25, 26].

The Fermi-Hubbard chain is given by an infinite chain of lattice sites labelled by integers $n$ with Hamiltonian:

$$\mathbf{H}_{\text{FH}} = -J\left[\sum_{n\sigma} \mathbf{c}_{n\sigma}^{\dagger} \mathbf{c}_{n+1\sigma} + \mathbf{c}_{n+1\sigma}^{\dagger} \mathbf{c}_{n\sigma}\right] + U\sum_{n} \mathbf{n}_{n\uparrow} \mathbf{n}_{n\downarrow}, \tag{G.1}$$

where $\mathbf{c}_{n\sigma}$ annihilates a fermion at site $n$ with spin $\sigma \in \{\uparrow, \downarrow\}$, $\mathbf{n}_{n\sigma} = \mathbf{c}_{n\sigma}^{\dagger}\mathbf{c}_{n\sigma}$, $J$ is the hopping parameter and $U < 0$ is the Hubbard interaction. There are basically two types of stable particles in the system: first, we have single particles. Second due to attractive Hubbard interaction we also have bound pairs of particles with opposite spin. We would like to study scattering of these particles at the following impurity:

$$\mathbf{V}(t) = \lambda\omega\big(\mathbf{n}_{0\uparrow} + \mathbf{n}_{0\downarrow}\big)\cos\omega t, \tag{G.2}$$

i.e. an extra chemical potential at site 0 that oscillates with frequency $\omega$ and driving strength $\lambda$ (in units of $\omega$). The scattering of single particles has already been studied in detail [53]. For bound pairs one can use standard methods of Floquet theory to derive an effective pair Hamiltonian in the limit $J \ll |U|, \omega$ as long as $|U|$ is not an integer multiple of $\omega$, i.e. the scattering is non-resonant [25]. The problem with resonant scattering is that it allows for a process called pair breaking, where a pair absorbs several energy quanta $\omega$ at the impurity and then breaks into two single particles. This makes resonant driving much harder to analyse. We will now show how to apply the above theorems to obtain analytical results in the resonant case.

## G.1 Properties of the Fermi-Hubbard chain without impurity

First, let us remind the reader of important properties of the plain Fermi-Hubbard chain for two particles of opposite spin. We can think of this system as a 2D lattice $|n, m\rangle = \mathbf{c}_{n\uparrow}^{\dagger}\mathbf{c}_{m\downarrow}^{\dagger}|\Omega\rangle$ (here $|\Omega\rangle$ denotes the vacuum), where the $n$ and $m$ are the position of the first and the second particle resp. The Hubbard interaction is then given by an extra interaction potential along the diagonal $n = m$. If one is interested in studying a bound pair formed by both particles it is natural to consider the regime $J \ll U$. In that limit the diagonal (describing a bound pair) will completely decouple from the rest of the system (describing two free particles). In order to derive the following results already in the form required later we will rescale $\alpha = J/\omega$ and $u = U/\omega$, even though $\omega$ is arbitrary for now. In the limit $\alpha \to 0$ while keeping $u$ constant, we can use the Schrieffer-Wolff transformation to block diagonalize the Hamiltonian $\mathbf{H}_{\text{FH}}$ [47,54]:

$$\frac{1}{\omega}\mathbf{H}_{\text{FH}} = \mathbf{W}_{\alpha}\Big(-|u|\tilde{\mathbf{P}} + \alpha^2\tilde{\mathbf{P}}\mathbf{h}_{\alpha}^{\tilde{\mathbf{P}}}\tilde{\mathbf{P}} + \alpha\tilde{\mathbf{Q}}\mathbf{h}_{\alpha}^{\tilde{\mathbf{Q}}}\tilde{\mathbf{Q}}\Big)\mathbf{W}_{\alpha}^{\dagger}, \tag{G.3}$$

where we defined $\tilde{\mathbf{P}}$ as the projectors onto the diagonal

$$\tilde{\mathbf{P}}|n, m\rangle = \begin{cases} |n, n\rangle, & n = m, \\ 0, & n \neq m, \end{cases} \tag{G.4}$$

and $\tilde{\mathbf{Q}} = \mathbf{1} - \tilde{\mathbf{P}}$ as the projector on the rest of the system. Note that the unitary $\mathbf{W}_{\alpha}$, which we neglected in the initial discussion (Section 2) as it only induces higher order corrections, is commonly denoted as $\mathbf{W}_{\alpha} = e^{-S_{\alpha}}$ and has the expansion

$$\mathbf{W}_{\alpha} = 1 + \frac{\alpha}{|u|}\tilde{\mathbf{Q}}\left[\sum_{n\sigma} \mathbf{c}_{n\sigma}^{\dagger}\mathbf{c}_{n+1\sigma} + \mathbf{c}_{n+1\sigma}^{\dagger}\mathbf{c}_{n\sigma}\right]\tilde{\mathbf{P}} + \mathcal{O}\big(\alpha^2\big). \tag{G.5}$$

For small $\alpha$ the Hamiltonian on $\tilde{\mathbf{P}}$ is given by

$$\mathbf{h}_\alpha^{\tilde{P}} = -\frac{2}{|u|}\left[\sum_n \eta_n^+ \eta_{n+1}^- + \eta_{n+1}^+ \eta_n^- - 2\right] + \mathcal{O}(\alpha),\tag{G.6}$$

where the pair creation/annihilation operators $\eta_n^+ = \mathbf{c}_{n\uparrow}^\dagger \mathbf{c}_{n\downarrow}^\dagger$ and $\eta_n^- = \mathbf{c}_{n\downarrow}\mathbf{c}_{n\uparrow}$. This effective pair Hamiltonian is a hopping Hamiltonian with (pair) hopping parameter $\frac{2J^2}{|U|}$ (in original units). For small $\alpha$, $\mathbf{h}_\alpha^{\tilde{Q}}$ becomes a 2D hopping Hamiltonian where the diagonal is excluded, however we will not need its exact form.

### G.2 Discussion of the Fermi-Hubbard chain with resonantly driven impurity

Now we also include the driven impurity $\mathbf{V}(t)$. In order to bring the system into a more convenient form perform the gauge transformation $e^{-i\lambda(\mathbf{n}_{0\uparrow}+\mathbf{n}_{0\downarrow})\sin\omega t}$ which gives [25]:

$$\frac{1}{\omega}\mathbf{H}^{\text{gauge}}(t) = -\frac{J}{\omega}\sum_{n\sigma}\left[g_n(\omega t)\mathbf{c}_{n\sigma}^\dagger \mathbf{c}_{n+1\sigma} + g_n^*(\omega t)\mathbf{c}_{n+1\sigma}^\dagger \mathbf{c}_{n\sigma}\right] + \frac{U}{\omega}\sum_n \mathbf{n}_{n\uparrow}\mathbf{n}_{n\downarrow},\tag{G.7}$$

with $g_n(\phi) = 1$ except for $g_{-1}(\phi) = e^{-i\lambda\sin(\phi)}$ and $g_0(\phi) = e^{i\lambda\sin(\phi)}$

The gauge transformation removed the driving term and replaced it by a time-dependent hopping from and to site 0. By subtracting the plain Fermi-Hubbard chain $\mathbf{H}_{\text{FH}}$ we find that the new impurity is given by:

$$J\mathbf{V}^{\text{gauge}}(t) = (\mathbf{H}^{\text{gauge}}(\omega t) - \mathbf{H}_{\text{FH}}) = -\sum_\sigma\left[(e^{-i\lambda\sin(\omega t)} - 1)\mathbf{c}_{-1\sigma}^\dagger \mathbf{c}_{0\sigma} + (e^{i\lambda\sin(\omega t)} - 1)\mathbf{c}_{0\sigma}^\dagger \mathbf{c}_{1\sigma} + \text{h.c.}\right],\tag{G.8}$$

or in Fourier components:

$$\mathbf{V}_a^{\text{gauge}} = \begin{cases} -J_a(\lambda)\left[\sum_\sigma \mathbf{c}_{-1\sigma}^\dagger \mathbf{c}_{0\sigma} + (-1)^a\mathbf{c}_{0\sigma}^\dagger \mathbf{c}_{-1\sigma} + (-1)^a\mathbf{c}_{0\sigma}^\dagger \mathbf{c}_{1\sigma} + \mathbf{c}_{1\sigma}^\dagger \mathbf{c}_{0\sigma}\right], & a \neq 0, \\ -(J_0(\lambda) - 1)\left[\sum_\sigma \mathbf{c}_{-1\sigma}^\dagger \mathbf{c}_{0\sigma} + \mathbf{c}_{0\sigma}^\dagger \mathbf{c}_{-1\sigma} + \mathbf{c}_{0\sigma}^\dagger \mathbf{c}_{1\sigma} + \mathbf{c}_{1\sigma}^\dagger \mathbf{c}_{0\sigma}\right], & a = 0. \end{cases}\tag{G.9}$$

Note that the impurity after the gauge transformation now acts on sites $n = -1, 0, 1$. Recall that the Schrieffer-Wolff transformation Eq. (G.3) still contains the unitary transformation $\mathbf{W}_\alpha$. In order to bring this Hamiltonian into the form Eq. (51) required for Theorem 3 we perform another unitary transformation:

$$\frac{1}{\omega}\mathbf{W}_\alpha^\dagger \mathbf{H}^{\text{gauge}}(t)\mathbf{W}_\alpha = -|u|\tilde{\mathbf{P}} + \alpha^2\tilde{\mathbf{P}}\mathbf{h}_\alpha^{\tilde{P}}\tilde{\mathbf{P}} + \alpha\tilde{\mathbf{Q}}\mathbf{h}_\alpha^{\tilde{Q}}\tilde{\mathbf{Q}} + \alpha\mathbf{W}_\alpha^\dagger \mathbf{V}^{\text{gauge}}(t)\mathbf{W}_\alpha.\tag{G.10}$$

We will now split $u = \frac{U}{\omega}$ into two contributions $u = u_0 + \alpha\delta u$, where $u_0$ plays the role of $A$ in Theorem 3 and $\delta u$ controls how $u$ approaches $u_0$ as $\alpha \to 0$. As a final step let us rescale time by $1/\alpha^2$ and add an energy offset:

$$\frac{1}{\alpha^2}\left[\frac{1}{\omega}\mathbf{W}_\alpha^\dagger \mathbf{H}^{\text{gauge}}(t/\alpha^2)\mathbf{W}_\alpha + |u|\right] = \tilde{\mathbf{P}}\mathbf{h}_\alpha^{\tilde{P}}\tilde{\mathbf{P}} + \frac{|u_0|}{\alpha^2}\tilde{\mathbf{Q}} + \frac{1}{\alpha}\tilde{\mathbf{Q}}(\mathbf{h}_\alpha^{\tilde{Q}} - \delta u)\tilde{\mathbf{Q}} + \frac{1}{\alpha}\mathbf{W}_\alpha^\dagger \mathbf{V}^{\text{gauge}}(t/\alpha^2)\mathbf{W}_\alpha.\tag{G.11}$$

This Hamiltonian is now of the form Eq. (40), where $A = |u_0|$ and in our units $\omega(\alpha) = 1/\alpha^2$, i.e. $\omega_0 = 1$. Furthermore one can easily check $\tilde{\mathbf{P}}\mathbf{V}^{\text{gauge}}(t)\tilde{\mathbf{P}} = 0$ and $\tilde{\mathbf{Q}}\mathbf{V}^{\text{gauge}}(t)\tilde{\mathbf{P}} \neq 0$.

Theorem 3 establishes that scattering from $\tilde{\mathbf{P}}$ to $\tilde{\mathbf{Q}}$ is suppressed in the limit $\alpha \to 0$, i.e. in the strong coupling limit the bound pair will not break at the resonantly driven impurity.

### G.3 Computation of the pair transmission

We complete the discussion of the bound pair scattering at the driven impurity by computing the pair transmission through the impurity using Eq. (52). For this we first need to determine the projectors $\mathbf{K}$ and $\mathbf{L}$:

The projector $\mathbf{K}$ is defined as the subspace on which

$$\mathbf{V}_0^{\text{eff}} = \tilde{\mathbf{P}}_0 \mathbf{V}_{0\leftarrow|u_0|} \mathbf{Q}_{|u_0|} \mathbf{G}_0^{\text{Q}} \mathbf{Q}_{|u_0|} \mathbf{V}_{|u_0|\leftarrow 0} \tilde{\mathbf{P}}_0 , \tag{G.12}$$

acts, i.e. the orthogonal complement of its kernel. Because $\mathbf{V}_{|u_0|\leftarrow 0}$ vanishes outside of sites $n = -1, 0, 1$ we only need to consider the three dimensional subspace of $\tilde{\mathbf{P}}_0$ spanned by

$$|-1,-1\rangle = \mathbf{c}_{-1\uparrow;0}^\dagger \mathbf{c}_{-1\downarrow;0}^\dagger |\Omega\rangle , \qquad |0,0\rangle = \mathbf{c}_{0\uparrow;0}^\dagger \mathbf{c}_{0\downarrow;0}^\dagger |\Omega\rangle , \qquad |1,1\rangle = \mathbf{c}_{1\uparrow;0}^\dagger \mathbf{c}_{1\downarrow;0}^\dagger |\Omega\rangle , \tag{G.13}$$

where $|\Omega\rangle$ denotes the vacuum and $\mathbf{c}_{n\sigma;a}^\dagger$ creates a particle at site $n$, with spin $\sigma$ in copy $a$. Similarly in $\mathbf{Q}_{|u_0|}$ we only need to consider the two dimensional subspace spanned by

$$\frac{|-1,0\rangle + |0,-1\rangle}{\sqrt{2}} = \frac{1}{\sqrt{2}} \Big( \mathbf{c}_{0\uparrow;|u_0|}^\dagger \mathbf{c}_{-1\downarrow;|u_0|}^\dagger |\Omega\rangle + \mathbf{c}_{-1\uparrow;|u_0|}^\dagger \mathbf{c}_{0\downarrow;|u_0|}^\dagger |\Omega\rangle \Big), \tag{G.14}$$

$$\frac{|0,1\rangle + |1,0\rangle}{\sqrt{2}} = \frac{1}{\sqrt{2}} \Big( \mathbf{c}_{1\uparrow;|u_0|}^\dagger \mathbf{c}_{0\downarrow;|u_0|}^\dagger |\Omega\rangle + \mathbf{c}_{0\uparrow;|u_0|}^\dagger \mathbf{c}_{1\downarrow;|u_0|}^\dagger |\Omega\rangle \Big). \tag{G.15}$$

In this basis $\mathbf{V}_{|u_0|\leftarrow 0}$ looks as follows for odd $|u_0|$:

$$\mathbf{V}_{|u_0|\leftarrow 0} = J_{|u_0|}(\lambda) \mathbf{A}_{\text{odd}} = J_{|u_0|}(\lambda) \sqrt{2} \begin{pmatrix} -1 & 1 & 0 \\ 0 & 1 & -1 \end{pmatrix} , \tag{G.16}$$

and for even $|u_0|$:

$$\mathbf{V}_{|u_0|\leftarrow 0} = -J_{|u_0|}(\lambda) \mathbf{A}_{\text{even}} = -J_{|u_0|}(\lambda) \sqrt{2} \begin{pmatrix} 1 & 1 & 0 \\ 0 & 1 & 1 \end{pmatrix} . \tag{G.17}$$

Let us define the vectors in $\tilde{\mathbf{P}}_0$

$$|a\rangle_{\text{odd}} = \frac{1}{\sqrt{2}} \begin{pmatrix} 1 \\ 0 \\ -1 \end{pmatrix} , \qquad |b\rangle_{\text{odd}} = \frac{1}{\sqrt{6}} \begin{pmatrix} 1 \\ -2 \\ 1 \end{pmatrix} , \qquad |c\rangle_{\text{odd}} = \frac{1}{\sqrt{3}} \begin{pmatrix} 1 \\ 1 \\ 1 \end{pmatrix} , \tag{G.18}$$

$$|a\rangle_{\text{even}} = \frac{1}{\sqrt{2}} \begin{pmatrix} 1 \\ 0 \\ -1 \end{pmatrix} , \qquad |b\rangle_{\text{even}} = \frac{1}{\sqrt{6}} \begin{pmatrix} 1 \\ 2 \\ 1 \end{pmatrix} , \qquad |c\rangle_{\text{even}} = \frac{1}{\sqrt{3}} \begin{pmatrix} 1 \\ -1 \\ 1 \end{pmatrix} , \tag{G.19}$$

which are chosen s.t. $\mathbf{A}|c\rangle = 0$, $|a\rangle$ is antisymmetric and $|b\rangle$ is the remaining orthogonal vector (we will drop the 'even'/'odd' index unless necessary). If we us the basis $|a\rangle, |b\rangle$ we can think of the effective potential Eq. (G.12) as a product of three two by two matrices (which, due to parity conservation, are diagonal). Unless one of the two diagonal elements of the Greens operator accidentally vanishes, $\mathbf{V}_0^{\text{eff}}$ neither sends $|a\rangle$ nor $|b\rangle$ to zero and thus we conclude $\mathbf{K} = |a\rangle\langle a| + |b\rangle\langle b|$. The remaining vector $|c\rangle$ defines the projector $\mathbf{L} = |c\rangle\langle c|$.

Therefore, the only non-zero matrix element of $\bar{\mathbf{K}} \mathbf{V}_0^{\text{eff}'} \bar{\mathbf{K}}$ is given by $\langle c| \bar{\mathbf{K}} \mathbf{V}_0^{\text{eff}'} \bar{\mathbf{K}} |c\rangle = \nu$. We can evaluate this using Eq. (52), which in this case simplifies to:

$$\bar{\mathbf{K}} \mathbf{V}_0^{\text{eff}'} \bar{\mathbf{K}} = \bar{\mathbf{K}} \mathbf{W}_0^{\dagger'} \mathbf{V}_0 \bar{\mathbf{K}} + \bar{\mathbf{K}} \mathbf{V}_0 \mathbf{W}_0' \bar{\mathbf{K}} + \sum_{b \neq |u_0|} \frac{\bar{\mathbf{K}} \mathbf{V}_{-b} \mathbf{V}_b \bar{\mathbf{K}}}{b - |u_0|} , \tag{G.20}$$

by writing all quantities as matrices (note that $\mathbf{W}'_0$ is also a symmetric hopping and thus, in this context, proportional to $\mathbf{A}_{\text{even}}$):

$$
\nu = \begin{cases} \frac{16}{3}\left[\sum_b \frac{J_{2b}(\lambda)^2}{2b-|u_0|} + \frac{1}{|u_0|}\right], & |u_0| \text{ odd}, \\ \frac{16}{3}\left[\sum_b \frac{J_{2b+1}(\lambda)^2}{2b+1-|u_0|}\right], & |u_0| \text{ even}. \end{cases} \tag{G.21}
$$

The second conclusion (ii) of Theorem 1 allows us to calculate $\langle c|\psi_0^+\rangle$. For that we need to calculate $\mathbf{G}_0^{\mathrm{P}} = \frac{1}{\varepsilon_0 - \mathbf{h}_0^{\tilde{\mathrm{P}}} + i\eta}$ with $\varepsilon_0 = -\frac{4}{|u_0|}(\cos k - 1)$ and $\mathbf{h}_0^{\tilde{\mathrm{P}}} = -\frac{2}{|u_0|}\left[\sum_n \eta_n^+ \eta_{n+1}^- + \eta_{n+1}^+ \eta_n^- - 2\right]$. Note that $\mathbf{h}_0^{\tilde{\mathrm{P}}}$ is equivalent to a tight-binding chain. The Greens function of a tight-binding chain is a standard result and can be evaluated using contour integration (see for example [48] (9.93) - (9.98)):

$$
\langle n,n|\mathbf{G}_0^{\mathrm{P}}|m,m\rangle = \frac{|u_0|e^{i|k||n-m|}}{4i\sin|k|}. \tag{G.22}
$$

The incoming state is given by a plane wave $\langle n,n|\phi\rangle = e^{ikn}$. By writing all appearing quantities in the conclusion (ii) of Theorem 1 as matrices and expressing them in the basis $|a\rangle, |b\rangle, |c\rangle$ one arrives after some computation at:

$$
\langle c|\psi_0^+\rangle = -\frac{4\sqrt{3}i\sin k}{6 + \left(3|u_0|\nu + (-1)^{|u_0|}8\right)e^{ik} + 2e^{2ik}}. \tag{G.23}
$$

Now we could use conclusion (iii) of Theorem 1 to calculate the full scattering wavefunction. However, there is a faster way to compute the transmission. Assume an incoming wave from the left, i.e. $k > 0$. Then we know that for $n > 1$

$$
\langle n,n|\psi_0^+\rangle = t_k e^{ikn}, \tag{G.24}
$$

where $t_k$ is the transmission amplitude. Therefore:

$$
t_k = e^{-ik}\langle 1,1|\psi_0^+\rangle = e^{-ik}\left[\langle 1,1|a\rangle\langle a|\psi_0^+\rangle + \langle 1,1|b\rangle\langle b|\psi_0^+\rangle + \langle 1,1|c\rangle\langle c|\psi_0^+\rangle\right] \tag{G.25}
$$

$$
= e^{-ik}\langle 1,1|c\rangle\langle c|\psi_0^+\rangle = \frac{e^{-ik}}{\sqrt{3}}\langle c|\psi_0^+\rangle = -\frac{4i\sin k}{6 + \left(3|u_0|\nu + (-1)^{|u_0|}8\right)e^{ik} + 2e^{2ik}}e^{-ik}. \tag{G.26}
$$

The transmission probability is finally given by $T_k = |t_k|^2$:

$$
T_k = \frac{1}{1 + \left(\frac{3|u_0|\nu + (-1)^{|u_0|}8 + 8\cos k}{4\sin k}\right)^2}. \tag{G.27}
$$

Note that this result does not depend on $\delta u$ which is a consequence of the fact that for $\alpha \to 0$ scattering in $\tilde{\mathrm{P}}$ is independent from the details in subspace $\tilde{\mathrm{Q}}$. Interestingly, Eq. (G.27) equals the result one would obtain by first studying the limit $\alpha \to 0$ for $|u|$ not close to an integer using the Floquet-Schrieffer-Wolff transformation (see [25, 26]) and then sending $|u| \to |u_0|$ to an integer. We give a plot of the pair transmission for $|u_0| = 1$ in Fig. 5. The general result shows similar features as the result obtained for non-resonant driving, especially the almost periodic structure that appears for large $\lambda$ (see discussion in [25, 26]). However, the most important difference is that for $\lambda \to 0$ the impurity does not become fully transparent, but instead shows finite transmission and reflection. Of course, if $\lambda \to 0$ before we send $\alpha \to 0$, then the impurity will be fully reflecting. But, in our case we first send $\alpha \to 0$ and only then consider the limit $\lambda \to 0$. This is again a manifestation of the fact that the two limits $\alpha \to 0$ and $\lambda \to 0$ are not interchangeable, like we already observed in the discussion of the toy model Section 3. Physically this means that our result applies to a regime $\alpha \ll \lambda$, which is out of reach of an ordinary perturbative expansion in driving strength $\lambda$.

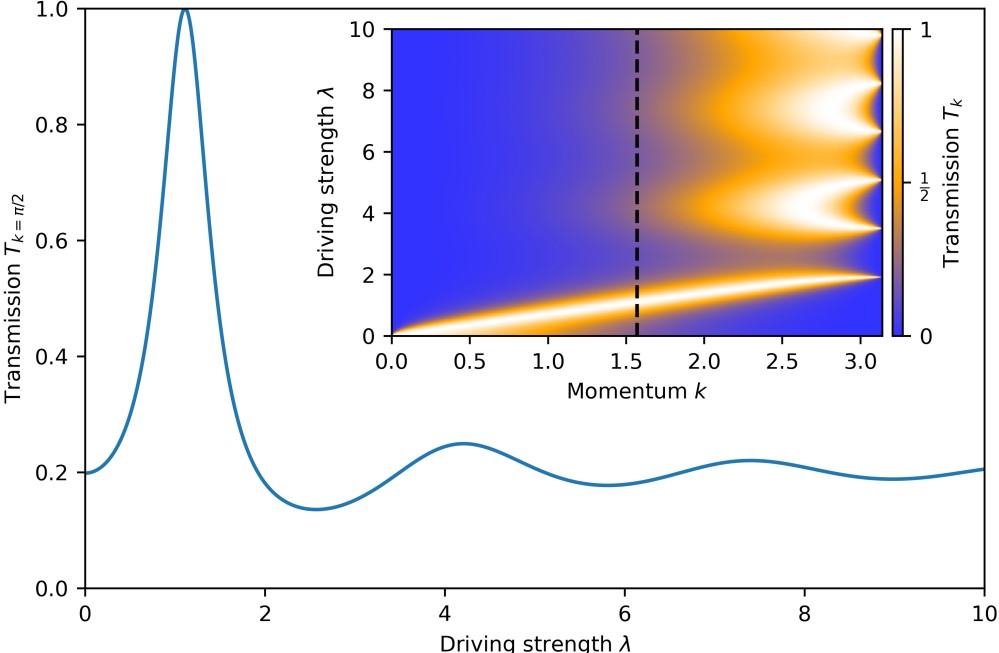

Figure 5: Pair transmission through the driven impurity for $u = u_0 = \frac{U}{\omega} = -1$ in the limit $\alpha = \frac{J}{\omega} \to 0$ as function of driving strength $\lambda$ for pair momentum $k = \pi/2$ (corresponding to the dashed line in the inset). The inset shows the transmission as function of both $k$ and $\lambda$.

## H  Application: Deep optical lattice with a resonantly driven impurity

In order to demonstrate the generality of Theorem 3 we would like to give another example: A resonantly driven impurity which couples the two lowest energy bands in a 1D optical lattice. Optical lattices are one of the most important platforms to simulate lattice systems in the lab (see for instance reviews [1, 9, 52]). They are generated by the standing wave of two counterpropagating laser beams. This induces an effective potential for the cold atoms described by a cosine potential [1, 30, 55]:

$$V_{\text{optical}}(x) = \frac{V_0}{2} \cos 2k_{\text{L}} x, \tag{H.1}$$

where $k_{\text{L}}$ is the wavenumber of the laser field. In the limit of a deep lattice $V_0 \to \pm\infty$, one can restrict the lowest energy states, which are the ground states $|j\rangle$ of each individual well $j$. The Hamiltonian on those states is given by a tight-binding chain:

$$\mathbf{H}_0 = -J \sum_j |j+1\rangle \langle j| + |j\rangle \langle j+1|, \tag{H.2}$$

where $J$ is the nearest neighbor hopping amplitude. Many applications of Floquet theory are based on this common approximation [1]. Of course the tight-binding approximation is just an approximation. One neglects higher energy bands. By introducing a resonantly driven impurity, with a driving frequency that is resonant between two bands one can excite particles from the lowest band to another band.[4] We will now demonstrate that Theorem 3 can also be applied to this scenario, even though the low-energy approximation is not generated by a Schrieffer-Wolff transformation.

---

[4]Resonant bulk driving has also been studied in optical lattices, see for instance [17].

The Schroedinger equation in potential Eq. (H.1) can be written in dimensionless form as the Mathieu equation [55]:

$$\partial_x^2 \psi(x) + (\varepsilon - 2v_0 \cos(2x))\psi(x) = 0, \tag{H.3}$$

where $v_0$ is the dimensionless potential depth and $\varepsilon$ is the dimensionless eigenenergy. We are interested in the deep optical lattice limit $v_0 \to \infty$. In this case the $n$'th band energy band ($n = 0, 1, 2, \ldots$) takes values in $\varepsilon \in [a_n, b_{n+1}]$, where asymptotically for $v_0 \to \infty$:

$$a_n \approx b_{n+1} = -2v_0 + 2(2n+1)\sqrt{v_0} - \frac{(2n+1)^2 + 1}{2^3} + \mathcal{O}(1/\sqrt{v_0}), \tag{H.4}$$

$$b_{n+1} - a_n \sim \sqrt{\frac{2}{\pi}} \frac{2^{4n+5}}{n!} v_0^{\frac{n}{2}+\frac{3}{4}} e^{-4\sqrt{v_0}}. \tag{H.5}$$

The lowest energy band can be described by a tight-binding Hamiltonian Eq. (H.2) where $J$ scales as follows [55]:

$$J \sim \frac{8\sqrt{2}}{\sqrt{\pi}} v_0^{\frac{3}{4}} e^{-4\sqrt{v_0}}. \tag{H.6}$$

## H.1 Scattering at the resonantly driven impurity

Let us now study the situation where a system is prepared in the lowest band. Then a driven impurity is added to this system which is resonantly driven with a frequency at resonance with the two lowest bands:

$$\omega = \frac{a_2 - a_1}{A} \approx \frac{4}{A}\sqrt{v_0}, \tag{H.7}$$

where $A = 1, 2, 3, \ldots$ is an integer. Since the system is initialized in the lowest band we would like to compare all energy-scales to the typical energy of the lowest band $J$. First, note that all high-energy bands $n = 2, 3, \ldots$ have an exponentially large energy in $J$ even in the extended Hilbert space and therefore we can safely neglect them and restrict to the first two bands $n = 0, 1$. Following the notation of this paper we call them $\tilde{P}$ ($n = 0$) and $\tilde{Q}$ ($n = 1$). Let us define $\alpha = \frac{1}{\sqrt{v_0}}$ which will play the role of the small parameter and write:

$$b_{n+1} - a_n = J \frac{16^n}{n!} \frac{1}{\alpha^n}. \tag{H.8}$$

In the deep lattice limit $\alpha \to 0$, the first two bands $n = 0$ and $n = 1$ show a clear separation of time-scales.

This allows us to study scattering at a driven impurity using Theorem 3. We will use an impurity $\frac{J}{\alpha}\mathbf{V}\cos\omega t$, where $\mathbf{V}$ for now is an arbitrary operator acting in a localized region in space.

The total Hamiltonian is now given by:

$$\mathbf{H}(t) = J\tilde{\mathbf{P}}\mathbf{h}^{\tilde{\mathbf{P}}}\tilde{\mathbf{P}} + A\omega\tilde{\mathbf{Q}} + \frac{J}{\alpha}\tilde{\mathbf{Q}}\mathbf{h}^{\tilde{\mathbf{P}}}\tilde{\mathbf{Q}} + \frac{J}{\alpha}\mathbf{V}\cos\omega t + (\text{higher bands}), \tag{H.9}$$

where we neglected the higher bands as discussed before and defined $\tilde{\mathbf{P}}$ and $\tilde{\mathbf{Q}}$ as projectors on the first and second band respectively. Note that their Hamiltonians are defined in such a way that $\mathbf{h}^{\tilde{\mathbf{P}}}$ and $\mathbf{h}^{\tilde{\mathbf{Q}}}$ remain finite as $\alpha \to 0$. Since $J$ depends on $\alpha$ let us rescale time:

$$\frac{1}{J}\mathbf{H}(t/J) = \tilde{\mathbf{P}}\mathbf{h}^{\tilde{\mathbf{P}}}\tilde{\mathbf{P}} + A\frac{\omega}{J}\tilde{\mathbf{Q}} + \frac{1}{\alpha}\tilde{\mathbf{Q}}\mathbf{h}^{\tilde{\mathbf{P}}}\tilde{\mathbf{Q}} + \frac{1}{\alpha}\mathbf{V}\cos\frac{\omega}{J}t. \tag{H.10}$$

This Hamiltonian is now in the form of Eq. (40), where

$$\omega(\alpha) = \frac{\omega}{J} \sim \frac{1}{\alpha^{\frac{5}{2}}} e^{\frac{4}{\alpha}} \gg \frac{1}{\alpha^2}. \tag{H.11}$$

In this case $\omega_0$ defined by Eq. (39) is formally $\omega_0 = \infty$, which however is compatible with Theorem 3.

We can now apply Theorem 3 on the Hamiltonian Eq. (H.10) and conclude that particles in the lowest band $n = 0$ will not get excited into the first band $n = 1$ by the resonantly driven impurity in the deep optical lattice limit $\alpha = \frac{1}{\sqrt{v_0}} \to 0$. Note that this result holds for any impurity operator $\mathbf{V}$.

## H.2 Computation of the transmission through the driven impurity

Next we can compute the resulting transmission inside the lowest band. First note that since $\omega_0 = \infty$ and the impurity does not contain a static part, Eq. (52) simply vanishes:

$$\bar{\mathbf{K}} \mathbf{V}_0^{\text{eff}'} \bar{\mathbf{K}} = 0. \tag{H.12}$$

Therefore, the projector $\mathbf{L} = 0$ is trivial. The wavefunction computed via conclusion (iii) from Theorem 1 becomes:

$$\mathbf{P} \left| \psi_0^+ \right\rangle = \left( 1 - \mathbf{G}_0^{\tilde{\mathbf{P}}} \mathbf{K} \frac{1}{\mathbf{K} \mathbf{G}_0^{\tilde{\mathbf{P}}} \mathbf{K}} \mathbf{K} \right) \left| \phi \right\rangle. \tag{H.13}$$

This result is independent of the details of the impurity potential, apart from the fact that it determines $\mathbf{K}$ as the space on which it acts.

Let us discuss an explicit example, where $\mathbf{V}$ is given by an external potential $V(x)$, which only acts on one site, say site $j = 0$. In terms of the tight-binding basis $|j\rangle$ this operator is given by $\mathbf{V} = v_{\text{imp}} |0\rangle \langle 0|$. Therefore $\mathbf{K} = |0\rangle \langle 0|$. Recall that the tight-binding Greens operator has the following explicit form Eq. (25) for an incoming plane wave $|\phi\rangle = \sum_j e^{ikj} |j\rangle$:

$$\langle j_1| \frac{1}{\varepsilon + \mathbf{h}^{\tilde{\mathbf{P}}} + i\eta} |j_2\rangle = \frac{e^{i|k||j_1 - j_2|}}{2i \sin |k|}, \qquad \qquad \varepsilon = -2 \cos k. \tag{H.14}$$

Using this we can compute:

$$\langle j| \mathbf{P} \left| \psi_0^+ \right\rangle = \langle j|\phi\rangle - e^{i|k||j|} \langle 0|\phi\rangle = e^{ikj} - e^{i|k||j|} = \begin{cases} 0, & jk \geq 0, \\ 2i \sin kj, & jk < 0. \end{cases} \tag{H.15}$$

This result means full reflection at the impurity since the transmission vanishes. Note that this can be understood similar to toy model, Section 3: since the wavefunction has to vanish at the location of the impurity, we have $\langle 0| \mathbf{P} \left| \psi_0^+ \right\rangle = 0$. This condition disconnects the two sides of the chain and therefore transmission is not possible anymore.

Note that the same impurity Eq. (H.10) driven non-resonantly, i.e. $A \notin \mathbb{Z}$, in the same regime $\alpha \to 0$ becomes fully transparent: since $\omega(\alpha)$ is exponentially large in $1/\alpha$, the system is well in the high-frequency regime. The leading term of the high-frequency expansion is just the time-averaged Hamiltonian [2]. Since the impurity does not contain a static part the impurity time average vanishes and therefore the impurity becomes fully transparent.

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
