# Peer review of "Suppression of scattering from slow to fast subsystems and application to resonantly Floquet-driven impurities in strongly interacting systems"

_SciPost Physics, doi:SciPost Phys. 16, 005 (2024)_

## Round 2 · Referee Report · Anonymous (Referee 1) · 2023-6-29

Report
This manuscript deals with scattering properties between subsystems with well separated time/energy scales. By considering a slow and a fast subsystem, the author rigorously proves an effective decoupling in the limit where the ratio alpha=T_fast/T_slow tends to zero. This implies, for example, that scattering from the slow subsystem to the fast one is suppressed as alpha -> 0. This result is proved in general by two theorems: the first states that, under the alpha -> 0 limit, the projection of the wave function into the space where the effective potential is nonzero vanishes, implying an 'artificial boundary condition' for the incoming particle in the slow subsystem. The second theorem uses this result to conclude that in this limit the projection of the wave function in the fast subsystem also vanishes. The results obtained are then illustrated in a Fermi-Hubbard chain, where the dynamics of pairs and individual particles are well separated. Since the associated energies for the two subsystems are not the same, that is, the energy bands do not overlap, the author includes a time periodic external potential, so that it induces scattering processes between the subsystems. The application of the theorem then shows that, at the separation limit of the time scales, pair breaking is suppressed.
In my opinion, the results obtained in this work are interesting and can be applied to a wide variety of systems. The manuscript is well written and its claims are rigorously verified. I also find the application of the toy model to introduce the terminology and elements of the scattering problem quite pedagogical. However, from the point of view of the novelty of the work, I am a little concerned with the way in which it was presented. Mainly the introduction shows a lack of background on a topic that is already known. For example, in the toy model discussed, it is possible to renormalize the energy at the 0_P site due to the presence of the Q chain. In the limit alpha --> 0, the P chain can be seen as a homogeneous chain with an impurity at its center, whose energy scales as 1/alpha. This can be considered as a high barrier that separates the system P into two parts. Therefore, it is not so surprising that in this limit the wave function at the 0_P site vanishes, and with it the probability of transmission to the other side of the chain.
Regarding the general acceptance criteria for SciPost Physics, I cannot recommend the manuscript in its current form. A revision of the introduction, claiming relevance to this topic, and including related bibliography, should be given for its consideration.
In my opinion, the results obtained in this work are interesting and can be applied to a wide variety of systems. The manuscript is well written and its claims are rigorously verified. I also find the application of the toy model to introduce the terminology and elements of the scattering problem quite pedagogical. However, from the point of view of the novelty of the work, I am a little concerned with the way in which it was presented. Mainly the introduction shows a lack of background on a topic that is already known. For example, in the toy model discussed, it is possible to renormalize the energy at the 0_P site due to the presence of the Q chain. In the limit alpha --> 0, the P chain can be seen as a homogeneous chain with an impurity at its center, whose energy scales as 1/alpha. This can be considered as a high barrier that separates the system P into two parts. Therefore, it is not so surprising that in this limit the wave function at the 0_P site vanishes, and with it the probability of transmission to the other side of the chain.
Regarding the general acceptance criteria for SciPost Physics, I cannot recommend the manuscript in its current form. A revision of the introduction, claiming relevance to this topic, and including related bibliography, should be given for its consideration.

---

## Round 2 · Referee Report · Anonymous (Referee 2) · 2023-7-10

Report
The author of this article derives theorems for systems composed of two subsystems that can scatter into each other where they can be described by the Lippmann-Schwinger equations. Furthermore, the theorems apply to the cases where one subsystem is much slower than the other. The interesting result is that for an initial incoming scattering state in the slow subsystem, its probability to scatter into the fast subsystem vanishes in the “extremely slow subsystem limit” . Furthermore, they apply their methodology to pair-breaking in a Fermi-Hubbard driven system and show analytical results.
I find the paper to be rather interesting and I am inclined to recommend it for publication. However, before that is done I would like the author to take into consideration the following comments and re-write the manuscript
1.) In the introduction, I think the author should cite more papers in order to relate the applicability of their methodology to the literature. For example, the author mentions that the method is especially useful in Floquet-Driven impurities which is good but a broader set of avenues must be thought of for its applicability (and cited). Please also cite the Floquet-theory papers, merely citing #[3] does not give the right credit so the author must do a thorough search of the literature.
2.) Related to available methodologies for solving driven-quantum systems, I think the authors should certainly mention/cite literature on the Floquet- Nonequilibirum Green Function techniques that have been covered in the plethora of papers like Phys. Rev. Lett. 113, 266801, Phys. Rev. B 87, 045402, Phys. Rev. Research 2, 033438, etc (& see cited papers within) that not only are nonperturbative and numerically exact but can also take into account effect of mixed states and quantum transport.
3.) Furthermore, I think the author may find their approach applicable to examples from the field of spintronics where there is distinction of the energy scales for classical or quantum localized spins that interact with electrons in a tight-binding chain. See for example Phys. Rev. Lett. 124, 197202
I think if the author can comment on points 1-3 in the introduction, manuscript will be stronger.
4.) Please add more examples from literature where this approach can be applicable so as to not stay restricted.
In introduction, “usual Schrieffer-Wolff regime” — please cite some reference so that the interested author can look into it. General readers may not know the “regime”. Furthermore, right after this, please introduce some story of what “pair-breaking” is before using it in the introduction (and put citations).
In the toy example, for the line “this means that particle on P will propagate slower by a factor of alpha ” — it is always better to describe that using a physical quantity, like momentum so that readers have something tangible to think of in terms of time-scale.
5.) For the equations, I recommend putting a comma or period after all of them as they are part of the manuscript writing as well as use Eq.(1) instead of (1).
6.) For the P and Q projection operators please write a mathematical expression in Section 2.2 so it is clear to the readers. Furthermore, what is the normalization constant for the state |phi> introduced in this section. Typically, that is \sqrt{N} where N is the “number of sites” in the TB chain.
Minor thing but for infinitesimal, it is better to use so as to avoid conflict with energy E = -2J cosk (they appear similar).
7.) Section 3 is written nicely where all the theorems are introduced, however for section 4.3, I encourage the authors to plot results in Eq. 59 and 60 and show what they look like, otherwise it does not give much insight. The plot should be able to show the importance of the methodology and provide interpretation as a function of k. Furthermore, please cut down on the equations in this section and move them to the appendix and focus only on the final result and what it teaches using your approach otherwise it is too difficult
to juggle with equations blocking the main message.
Lastly, I think the authors should also point out to other examples in the literature that can be solved using this approach as a future work. I would like to see the above changes implemented in the manuscript.
I find the paper to be rather interesting and I am inclined to recommend it for publication. However, before that is done I would like the author to take into consideration the following comments and re-write the manuscript
1.) In the introduction, I think the author should cite more papers in order to relate the applicability of their methodology to the literature. For example, the author mentions that the method is especially useful in Floquet-Driven impurities which is good but a broader set of avenues must be thought of for its applicability (and cited). Please also cite the Floquet-theory papers, merely citing #[3] does not give the right credit so the author must do a thorough search of the literature.
2.) Related to available methodologies for solving driven-quantum systems, I think the authors should certainly mention/cite literature on the Floquet- Nonequilibirum Green Function techniques that have been covered in the plethora of papers like Phys. Rev. Lett. 113, 266801, Phys. Rev. B 87, 045402, Phys. Rev. Research 2, 033438, etc (& see cited papers within) that not only are nonperturbative and numerically exact but can also take into account effect of mixed states and quantum transport.
3.) Furthermore, I think the author may find their approach applicable to examples from the field of spintronics where there is distinction of the energy scales for classical or quantum localized spins that interact with electrons in a tight-binding chain. See for example Phys. Rev. Lett. 124, 197202
I think if the author can comment on points 1-3 in the introduction, manuscript will be stronger.
4.) Please add more examples from literature where this approach can be applicable so as to not stay restricted.
In introduction, “usual Schrieffer-Wolff regime” — please cite some reference so that the interested author can look into it. General readers may not know the “regime”. Furthermore, right after this, please introduce some story of what “pair-breaking” is before using it in the introduction (and put citations).
In the toy example, for the line “this means that particle on P will propagate slower by a factor of alpha ” — it is always better to describe that using a physical quantity, like momentum so that readers have something tangible to think of in terms of time-scale.
5.) For the equations, I recommend putting a comma or period after all of them as they are part of the manuscript writing as well as use Eq.(1) instead of (1).
6.) For the P and Q projection operators please write a mathematical expression in Section 2.2 so it is clear to the readers. Furthermore, what is the normalization constant for the state |phi> introduced in this section. Typically, that is \sqrt{N} where N is the “number of sites” in the TB chain.
Minor thing but for infinitesimal, it is better to use so as to avoid conflict with energy E = -2J cosk (they appear similar).
7.) Section 3 is written nicely where all the theorems are introduced, however for section 4.3, I encourage the authors to plot results in Eq. 59 and 60 and show what they look like, otherwise it does not give much insight. The plot should be able to show the importance of the methodology and provide interpretation as a function of k. Furthermore, please cut down on the equations in this section and move them to the appendix and focus only on the final result and what it teaches using your approach otherwise it is too difficult
to juggle with equations blocking the main message.
Lastly, I think the authors should also point out to other examples in the literature that can be solved using this approach as a future work. I would like to see the above changes implemented in the manuscript.

---

## Round 2 · Referee Report · Anonymous (Referee 3) · 2023-7-18

Strengths
The manuscript seems to be mathematically rigorous. The derivation is rather detailed.
Weaknesses
The main weakness is that the author merely derives a formal relation for a quite artificial limit. There is no physical insight beyond that.
Report
The authors considers scattering between two subsystems with very different
time scales and proofs two theorems that hold in the limit in which the
ratio of these time scales tends to zero. The theorems concern formal
relations obeyed by intermediate expressions from scattering theory. The
result is also formulated for the corresponding expression for an ac-driven
impurity.
While the manuscript seems technically correct, it lacks physical
motivation and finally presents expressions that may or may not be useful
for solving any specific problem. Moreover, to me the limit considered
looks rather artificial and does not provide physical insight.
In conclusion, I cannot recommend the manuscript for publication, at least
not in the sections AMOP or Condensed Matter. I do not want to judge
whether it could fit to a more mathematically oriented journal or the
corresponding section of SciPost.
time scales and proofs two theorems that hold in the limit in which the
ratio of these time scales tends to zero. The theorems concern formal
relations obeyed by intermediate expressions from scattering theory. The
result is also formulated for the corresponding expression for an ac-driven
impurity.
While the manuscript seems technically correct, it lacks physical
motivation and finally presents expressions that may or may not be useful
for solving any specific problem. Moreover, to me the limit considered
looks rather artificial and does not provide physical insight.
In conclusion, I cannot recommend the manuscript for publication, at least
not in the sections AMOP or Condensed Matter. I do not want to judge
whether it could fit to a more mathematically oriented journal or the
corresponding section of SciPost.
Requested changes
None

---

## Round 3 · Referee Report · Anonymous (Referee 1) · 2023-9-24

Report

After reading the author's response to the comments of the referees, along with the revisions made to the manuscript, I think the physical motivation is now clarified. As this issue has been properly addressed in this revised version, I recommend its publication in SciPost Physics.

---

## Round 3 · Referee Report · Anonymous (Referee 3) · 2023-10-11

Strengths

see first report

Weaknesses

see first report

Report

I acknowledge that as compared to the former version, the motivation and the presentation have been improved. However, I still have reservations concerning the importance of the work and, thus, the fulfillment of the acceptance criteria at SciPost.

As already expressed in my first report, the manuscript deals with a somewhat artificial limit which seems far from any relevant physical situation. Thus, at least from the perspective of solid-state physics, I do not judge it as groundbreaking or opening any new direction. Since the manuscript is technically at a high level, it should nevertheless merit publication at some place, e.g. on a more mathematically oriented platform.
  • validity: top
  • significance: ok
  • originality: good
  • clarity: good
  • formatting: good
  • grammar: -

Author:  Friedrich Hübner  on 2023-11-29  [id 4156]

(in reply to Report 2 on 2023-10-11)

Dear referee,

thank you very much for your review and for acknowledging the high technical level of my work. I would like to add a short comment about the ‘artificial limit’: I fully agree that my results only apply in a certain limit, in particular only if the model shows the specific relative scaling of band widths as described in section 2. In appendices G and H I discuss how my method can be applied to study driven impurities in two systems, the Hubbard chain and the optical lattice, both of which are canonical models in solid-state physics. It is also common to approximate these models by their low-energy effective theories and the resulting models (Heisenberg chain, tight-binding chain) are well-studied models of solid-state physics. Most notably, these low-energy effective theories show the correct scaling required for my method to work. Furthermore, the same scaling emerges naturally out of the Schrieffer-Wolff transformation, which is a common technique to derive effective Hamiltonians in bands (see section 2).

Therefore, in my opinion, the scaling is not an ‘artificial limit’, but rather appears naturally in many solid-state models.

---

## Round 3 · Referee Report · Anonymous (Referee 2) · 2023-10-15

Report

The author has now revised the paper addressing most of the concerns from my previous report and added more examples. I think the paper may be published.

However, I urge the author to expand the citations being made regarding Floquet NEGF as I have highlighted in my previous report. Just three papers [39-41] are currently cited for Floquet NEGF that do not do just justice to the field. Please refer to my previous report and use the mentioned papers to extensively cite the literature.

One other objection I have is related to the following line in the new introduction:
"If suitably established this perturbative expansion would also add another tool to the toolbox of Floquet theory, which would be useful to
describe impurities which are too complicated to study via non-perturbative methods (like
the non-equilibrium Green’s function method [39–41])"

It is not clear to me what author means by "too-complicated to study via non-perturbative methods", as far as my knowledge in the field, I do not think impurities are at all "complicated" to be studied via NEGF methods. Please omit this line as it is inaccurate. One could essentially say that the technique presented can complement other techniques (as an example).

My final recommendation for the revised manuscript is that it can be published in SciPost but the above corrections must be implemented before that happens.

---

## Round 3 · Author Response

Warnings issued while processing user-supplied markup:

  • Inconsistency: Markdown and reStructuredText syntaxes are mixed. Markdown will be used.
    Add "#coerce:reST" or "#coerce:plain" as the first line of your text to force reStructuredText or no markup.
    You may also contact the helpdesk if the formatting is incorrect and you are unable to edit your text.

Dear editor,

I would like to resubmit my manuscript to SciPost Physics. In the following I will first broadly motivate the changes that I made to the manuscript and explain why I believe it is suitable for SciPost Physics. Then I will give an individual response to each referee. In the end you find a list of changes.

1. Discussion of the revision and contributions of the manuscript

In general, the referees agreed that the results of the paper are both very general and mathematically justified. It was indeed my original intention to write a paper which focuses on the mathematical derivation of result. But, I can fully understand that the referees were criticizing the lack of physical motivation for the physical situation I study in the paper.

Therefore, I decided to focus the paper on Floquet theory, which also was the original motivation for the project: In physics we often simplify complicated physical systems by restricting to their lowest energy band(s). Examples for this are the tight-binding approximation in lattice systems or the Schrieffer-Wolff transformation in strongly interacting systems. If the energy barrier to the next band is sufficiently high, this low-energy subsystem will stay intact even in the presence of perturbations. However, nowadays, in the context of Floquet-engineering one tries to use externally driven systems to generate the desired physical effects. One of the popular methods is to use high-frequency driving, where a) the system can be well-described by an effective static Floquet Hamiltonian and b) heating is suppressed. In many theoretical works one starts directly from the low-energy description of the system and then adds a high-frequency Floquet drive to achieve desired effects (see for instance Rev. Mod. Phys. 89, 011004 or https://doi.org/10.1080/00018732.2015.1055918). This approximation is justified, as long as the driving frequency is not at resonance with the energy barrier between the lowest band and a higher band. If such a resonance occurs then the higher band cannot longer be neglected, since particles will be continuously pumped from the lowest band to the higher band. As has been studied before, if the resonant driving is applied to the whole system, then both bands hybridize and effectively the low-energy description is destroyed (see for instance PhysRevLett.116.125301 and also my discussion around equation (8)).

In my work I study what happens if instead of applying the driving to the whole system, one only applies driving in a localized region, i.e. a resonantly driven impurity. The surprising result is that for a driven impurity, particles do not get excited into the higher band. Actually these excitations are suppressed in the same regime where bulk driving excites all the particles. The physical reason for this is that typically emergent low-energy descriptions show a clear separation of time-scales compared to the higher energy bands, i.e. dynamics in the low-energy band are much slower than dynamics of particles in the high-energy bands. This separation of time-scales prohibits an efficient coupling between both bands.

With this motivation in mind I decided to restructure the paper: I have completely rewritten the introduction and I also added a motivation section where I introduce the physical situation step by step. Then (Section 3 and 4) I discuss the static toy model and derive the general result for static impurities as before (Section 3 and 4 are mostly unchanged). After that I have added a new section, where I derive a general statement for Floquet-driven impurities as well (Section 5). In this section I present the main ideas that were previously in the 'bound pair at Floquet-driven impurity' section and apply them to a general situation (following the proposal of referee 2 to make this section more clear). The remaining parts of the discussion of the bound pair example where moved to the appendix. In the appendix I apply the results to two examples. First, the bound pair in the Fermi-Hubbard chain, which is a simple example for a low-energy system described by a Schrieffer-Wolff transformation. Then I added a second example: a deep optical lattice, whose emergent low-energy description is the tight-binding chain. The driving frequency of the impurity is at resonance with the energy difference between the lowest band and the next highest band. This example is both physically interesting, as the tight-binding approximation is the basis for simulating and manipulating lattice systems in optical lattices, as well as mathematically interesting, since the relative scalings of terms are quite different from the Schrieffer-Wolff regime. Still, my result applies and predicts that excitations into the higher band are suppressed.

I hope that this convinces you and the referees that the physical situation I study in this paper is not an 'artificial limit' (as suggested by referee 3), but instead it is a natural description for situations, when a low-energy band is resonantly coupled to a high-energy band by a driven impurity. I believe that this work satisfies the acceptance criteria for SciPost Physics, in particular:

Detail a groundbreaking theoretical/experimental/computational discovery: To best of my knowledge the resonant coupling of low-energy bands to high-energy bands via a driven impurity has not been systematically studied before. The results are groundbreaking in the sense that a) the suppression of excitations between both bands seems unintuitive and b) the effect of a resonantly driven impurity is much different from resonant bulk driving. Both conclusions were surprising to other senior researchers in the field. In my opinion there is also another important discovery in this work: It establishes c) a new kind of approximation that allows to derive analytical results for scattering at resonantly driven impurities. Most importantly, the approximation is not based on weak coupling, like the Born approximation, but instead it is based on a separation of time-scales. These separation of time-scales ideas are at the heart of the common approximations in Floquet-theory, like the high-frequency expansion or the Floquet-Schrieffer-Wolff transformation. In this regard, one could view the results of this paper as extensions of these methods to resonantly driven impurities.

Open a new pathway in an existing or a new research direction, with clear potential for multipronged follow-up work: While most of the groundbreaking theoretical and experimental work with Floquet systems so far focused on bulk driven systems, there are also proposals to use driven impurities to manipulate quantum systems. I can easily imagine that driven impurities may become important parts of quantum simulators or quantum computers (or at least good descriptions for Floquet driving which is applied only to a local region). Resonant driving in particular, for instance could be used to actively pump particles from one energy band to another. For applications like that the suppression of excitations as derived in this paper has to be taken into account. If driven impurities become important experimental devices, there will also be a need for efficient analytical tools to predict their behaviour. The results of this work already allow to gain insights into the effects the impurity has on the quantum system in case the driving is at resonance with some other band. So far, they apply only to systems with a clear separation of time-scales (equivalently with significantly different bandwidths). However, the results are based on Taylor expansions, which indicates that it should be possible to derive a systematic perturbative expansion order by order in $\alpha$ (the ratio of time-scales or bandwidths). If such an expansion is properly established it would allow to also predict the behaviour of resonant impurity driving in case where $\alpha$ is finite. As such it could become an important tool similar to the high-frequency expansion to study resonance effects at driven impurities (in particular since it is non-perturbative in coupling strength).

Another interesting direction of research would be to experimentally realize resonantly driven impurities as described in the beginning of this letter and to measure the suppression of excitations. I can in particular imagine a realization of the deep optical lattice example, since optical lattices are well-established devices in cold atom physics.

Responses to the referees

Referee 1:

Dear referee 1, thank you for your comments. I hope the revised introduction and the new motivation section address your concerns about the lack of background.

Regarding the renormalization of energy in the toy model: Indeed, one way of thinking about this system is to integrate out system Q and have an effective (or renormalized) impurity for system P. This is in fact what is done in the treatment of the general case, see Eq. (32). As it turns out this effective impurity is scaled by $1/\alpha$, i.e. it is very strong. At this point one can draw the analogy to scattering at a high potential localized at $\ket{0_P}$, where one of course expects that all particles are reflected, as tunneling through the energy barrier is strongly suppressed. Mathematically both situations are very similar and indeed give rise to the same result.

However, physically this effective impurity is not an impenetrable potential, but instead an accessible gateway to another physical system. Therefore, the impurity is not strong in the sense of a high potential, but in the sense of having a large hopping between from chain P to chain Q. In this case, intuitively, the particles should have a high chance to actually switch chains at the impurity. This is not the case, which is, in my opinion, surprising.

Referee 2:

Dear referee 2, thank you for your comments. As described in the general response, I reworked the manuscript and focused it more on Floquet-theory, taking into account your comments. Let me go through them one by one:

  1. I completely revised the introduction. Now I am focusing on resonantly Floquet-driven impurities, where P is a slow low energy band and Q is a fast high energy band. Coupling them with a resonantly driven impurity will allow scattering between them. If the band P is emerging from some low-energy description, then there is often a clear separation of time-scales between P and Q. This happens in the Schrieffer-Wolff transformation on which the pair breaking example is based. As another example, I added the scattering of particles in the lowest band of an optical lattice, coupled to the next energy band.

While the result is technically applicable to static systems as well, I do not expect it to be relevant outside of Floquet-theory, simply because a separation of time-scales is typically accompanied by a separation of energy-scales, which prevents any scattering between both systems. This might change, in case one is able to extend the result into a full perturbative series. This would allow to go beyond the approximation of a strong separation of time-scales and also study impurities which couple bands with comparable bandwidths. This could then be applied both to static and driven impurities.

  1. I agree that I should mention the non-equilibrium Greens function method. I added references in the introduction.

  2. Thank you very much for pointing out the field of spintronics. Indeed, these systems show a separation of time-scales. Therefore, if a system like that is coupled with a resonantly driven impurity, the techniques of the paper will be applicable.

  3. I added a motivation section, where I briefly give an overview over the Schrieffer-Wolff transformation and its extension to Floquet theory.

  4. I checked the equations again and inserted commas and periods, where I had missed them.

  5. I added the explicit definition of the projectors for the toy model. The normalization of $\ket{\phi}$ is not important since the Lippmann-Schwinger equations are linear: Multiplying $\ket{\phi}$ by some factor will simply multiply the resulting scattering state by the same factor. Note however, that the number of sites of the tight-binding chain is infinite. The normalization factor to obtain orthonormal states is $1/\sqrt{2\pi}$. I also changed the $+i\epsilon$ to $+i\eta$ (I could not see which letter you were proposing to replace the $\epsilon$ on the scipost webpage, so I decided to use $\eta$, since it is frequently used).

  6. I replaced Section 4 by a general section, where I apply the results from the previous section to driven impurities and give another theorem for driven impurities. The technical details of the specific model were moved to the appendix. This should make this section more clear and in particular demonstrate the generality of the result. For the pair transmission I also added a plot of the transmission in the appendix and a short discussion of it.

Referee 3

Dear referee 3, thank you for your comments. I fully agree that the previous version of the manuscript lacked motivation. Therefore I decided, as described in the general response, to completely rewrite the introduction and focus more on Floquet-theory. While for static systems the considered limit is indeed somewhat artificial, it appears naturally when one couples an emergent low energy subspace with the high-energy theory by a resonantly driven impurity. In such cases, often there is a separation of time-scales (or equivalently a separation of bandwidths) between both systems (see the motivation section and also at the two examples that I give).

In my opinion the main insight of this paper is that the separation of time-scales prohibits an efficient coupling between both systems via an impurity. I hope the arguments I gave in the general response convince you that the physical situation is not just an 'artificial limit' but instead is a common feature of low-energy descriptions of systems. I have also included an additional example of a deep optical lattice in Appendix H to further emphasize the applicability of the result.

---

## Round 3 · List of Changes

Abstract completely rewritten

Introduction completely rewritten

Section 2 (Motivation) added

Section 3 (Toy model), formerly Section 2:

# First sentence changed
# Added parenthesis in third sentence ('for instance, the typical group velocity ...')
# Sentence added ('This resembles the effect of the resonant driving ...')
# Added definition of projectors P and Q, Eq. (17) and (18)

Section 4 (Static case), formerly Section 3:

# First sentence changed
# Last sentence of 4.1 changed ('In fact, in Appendix G ...')
# Final paragraph of 4.1 (former 3.1) removed

Section 5 (Driven case) added

Conclusion completely rewritten

Appendix E (Technical assumptions)

# removed the parenthesis in both the third to last and last sentence

Appendix F (Derivation of (52)) added

Appendix G (Bound pairs) added

Appendix H (Deep optical lattice) added

In general former Section 4 and former Appendix F were restructured and distributed over Section 2, Section 5, Appendix F and Appendix G.

---

## Round 5 · Author Response

I took care of the minor revisions asked for by the referee. I added more citations about non-equilibrium Green's function methods. I have also clarified the sentence '... which would be useful to describe impurities which are too complicated to study via non-perturbative methods ...'. and replaced `too complicated' by `too computationally expensive'.
About this I would like to add the following comment: Of course one can in principle study any impurity in any model using NEGF or other non-perturbative methods. What I meant with 'too complicated' was that for sufficiently complicated impurities, performing analytical or numerical computations using exact non-perturbative methods might become infeasible given the available resources. I am not an expert in the theory of NEGF, but to my understanding, in practice, if one would like to apply NEGF to impurity scattering it seems very crucial that the self-energy of the leads is known. For the standard tight-binding lattices without particle interaction these are of course well known.
Please consider the two examples I discuss in appendix G and H. In appendix G I discuss an impurity where the leads are given by semi-infinite Hubbard chains (so particles in the leads are interacting). In appendix H the leads are given by semi-infinite optical lattices (particles are non-interacting, but this model contains infinitely many bands). I do not think the full non-perturbative self-energies of both models are known.
For more general impurities, especially if they are embedded in interacting systems in higher dimensions, an exact numerical analysis using non-perturbative methods might become infeasible due to the exponentially large Hilbert space. This is of course a general problem in interacting many-body quantum systems and there have been many ideas on how to still make good predictions. One of them is the idea of separation of scales, which is the basis for my analysis (but also behind other well-known techniques like the high-frequency expansion or hydrodynamic gradient expansions): The two above mentioned examples can be solved analytically in the regime I am considering, which demonstrates the drastic simplification compared to exact non-perturbative methods. Due to the completely general derivation my technique can also be applied to more general models (in particular interacting ones) given that they are in the correct regime. Even if the resulting simplified expressions are not analytically solvable, they can still add physical intuition and systematically pinpoint the most relevant objects to be computed via other methods. In this paper I only discussed the zeroth order approximation, but by extending to higher order one can systematically gain insight into resonant scattering at impurities even if the system is only approximately in that regime.
So to conclude, in my opinion the method I developed in this paper can help to gain understanding of impurities in physical situations which are difficult to treat with exact non-perturbative methods due to finite availability of resources.
Kind regards,
Friedrich Huebner

---

## Round 5 · List of Changes

non-perturbative methods (like the non-equilibrium Green’s function methods [21,39–45]) in
studying scattering at impurities and could perhaps be used to gain insights into impurities
that are too computationally expensive to treat in practice via these methods.'

---

## Editorial Decision

published